# Multi-Round Human–AI Collaboration with User-Specified Requirements

Sima Noorani [* 1]   Shayan Kiyani [* 1]   Hamed Hassani [1]   George Pappas [1]

## Abstract

As humans increasingly rely on multi-round conversational AI for high-stakes decisions, principled frameworks are needed to ensure such interactions reliably improve decision quality. We adopt a human-centric view governed by two principles: counterfactual harm, ensuring the AI does not undermine human strengths, and complementarity, ensuring it adds value where the human is prone to err. We formalize these concepts via user-defined rules, allowing users to specify exactly what harm and complementarity mean for their specific task. We then introduce an online, distribution-free algorithm with finite-sample guarantees that enforces the user-specified constraints over the collaboration dynamics. We evaluate our framework across two interactive settings: LLM-simulated collaboration on a medical diagnostic task and a human crowdsourcing study on a pictorial reasoning task. We show that our online procedure maintains prescribed counterfactual-harm and complementarity violation rates even under non-stationary interaction dynamics. Moreover, tightening or loosening these constraints produces predictable shifts in downstream human accuracy, confirming that the two principles serve as practical levers for steering multi-round collaboration toward better decision quality without the need to model or constrain human behavior.

## 1. Introduction

Conversational engagement with AI systems is already happening at massive scale, with humans increasingly relying on multi-round interactions to guide real-world decisions in domains ranging from medical triage and legal research to personal finance and everyday planning. In these settings, the AI participates in an evolving dialogue, refining information over multiple rounds as the human asks follow-up questions, challenges earlier outputs, or requests clarification. Despite the prevalence of this mode of interaction, we still lack principled ways to evaluate whether such collaborations are working well, or to design them so that they reliably improve decision quality under uncertainty.

Historically, both classical Aumann (1976) and recent work Collina et al. (2025) has often conceptualized collaboration through the lens of agreement: multiple agents interact in order to reconcile beliefs, converge to a shared posterior, or reach consensus. While this perspective has yielded deep theoretical insights, it implicitly treats the human and the AI as symmetric agents with comparable authority, responsibility, and incentives. In practice, however, this symmetry assumption is fundamentally misaligned with how conversational AI systems are deployed and used. Humans, not AI systems, are ultimately accountable for decisions and their consequences. Moreover, engagement with an AI system is itself a human choice: the human decides whether to consult the AI, how long to continue the interaction, how much trust to place in its responses, and when to disengage altogether.

Motivated by this asymmetry, we adopt a human-centric view of collaboration. We ask how a human–AI interaction should be designed to incentivize a human to engage in it in the first place. From this perspective, a successful human–AI collaboration must satisfy two minimal requirements. First, interaction with the AI should not undermine the human's existing strengths, a property we refer to as *counterfactual harm*. If the human is already poised to make a correct judgment, engaging with the AI should not degrade that outcome. Second, the collaboration should create genuine value precisely in situations where the human is likely to err, by helping recover correct outcomes that the human would otherwise miss, a property we refer to as *complementarity*.

In this work, we introduce a carefully designed algorithmic framework for multi-round human–AI collaboration under uncertainty. Our framework allows for an arbitrary flow of information between the human and the AI through natural language interaction, while governing the collaboration dynamics in a principled manner so as to satisfy the two requirements of counterfactual harm and complementarity.

---

[1]Department of Electrical and Systems Engineering, University of Pennsylvania, USA. Correspondence to: Sima Noorani <nooranis@seas.upenn.edu>.

*Proceedings of the 43$^{rd}$ International Conference on Machine Learning*, Seoul, South Korea. PMLR 306, 2026. Copyright 2026 by the author(s).

Along the way, our contributions span three fronts.

*(i) Problem formulation and modeling.* In Section 3, we propose two key modeling contributions. First, we introduce an interaction protocol in which a human and an AI collaborate on solving one problem at a time through a multi-round dialogue. This model captures the sequential nature of conversational interaction while providing an explicit algorithmic handle over the AI's uncertainty quantification, which can be shaped to enforce different collaboration constraints. Second, we develop a general framework for specifying constraints on collaboration dynamics through user-defined indicator functions of counterfactual harm and complementarity. These indicator functions allow a wide range of meaningful, application-specific notions of safe and productive collaboration to be translated into concrete constraints on how the AI communicates uncertainty.

*(ii) Algorithms and guarantees.* In Section 4, we develop a finite-sample, online calibration algorithm that can provably enforce user-defined collaboration constraints over time. Our guarantees hold without any assumptions on the distribution of problems, the human's behavior, or the internal mechanisms of the AI system. Despite this generality, the algorithm ensures that the AI's uncertainty communication adapts online so as to satisfy counterfactual harm and complementarity constraints throughout the interaction.

*(iii) Extensive experiments.* In Section 6, we provide an extensive empirical evaluation across two complementary interactive settings: LLM-simulated collaboration at scale and a real human crowdsourcing study. We (a) empirically validate that the online procedure maintains the prescribed counterfactual-harm and complementarity violation rates over time even under non-stationary interaction dynamics, and (b) show that tightening or loosening these targets produces predictable shifts in downstream human decision quality, confirming that the two principles act as practical levers for steering multi-round collaboration.

## 2. Related Works

In this section, we only focus on closely related literature, and provide an expanded discussion in Appendix B.

**Human-AI Collaboration through Prediction Sets.** A growing line of work studies prediction sets as interfaces for human decision support (Straitouri et al., 2023; 2024; Hullman et al., 2025; Babbar et al., 2022; Toni et al., 2024). These works generally treat the AI's output as a static hint and consider only single-round interactions, leaving open the question of how to handle iterative refinement where a human may update beliefs or revise proposals across multiple rounds. Our framework builds upon Noorani et al. (2025b), which provides the conceptual foundation for formalizing collaboration through prediction sets via counter-

factual harm and complementarity. We extend this single-round foundation to a multi-round formulation, allowing the AI's uncertainty sets to adapt dynamically to the evolving conversational transcript. This enables the principled control of collaboration constraints throughout the entire dialogue rather than at a single point in time.

**Theoretical Frameworks for Human-AI Synergy** A broader literature seeks to formalize complementarity, where the joint system outperforms either agent alone. One line models this using Bayesian frameworks or taxonomies of collaboration modes (Bansal et al., 2021; Steyvers et al., 2022; Rastogi et al., 2023). Another analyzes how algorithmic outputs shape human decisions through strategic influence or noisy ranking models (Cowgill & Stevenson, 2020; Kleinberg & Raghavan, 2021; Donahue et al., 2024). Our framework departs from these approaches in three ways: (1) we treat the human as a complete black box, requiring no assumptions on behavior; (2) our guarantees are distribution-free and remain valid under arbitrary shifts in human behavior; and (3) we address multi-round dialogue rather than single-stage interactions.

**Theoretical and Algorithmic Multi-Round Collaboration.** Multi-round collaboration is largely studied through agreement protocols aimed at reaching consensus or aggregating information (Aumann, 1976; Aaronson, 2004; Collina et al.; 2025; Geanakoplos & Polemarchakis, 1982). Our framework differs in its assumptions: while these protocols require agents to communicate probabilistic beliefs or best-response action, we treat the human as a black box who only provides a prediction set. Furthermore, while agreement protocols place the human and the AI on equal footing, we adopt a human-centered point of view. This ensures that the AI remains harmless and helpful to the human, regardless of whether the parties ever reach an agreement.

## 3. Background: Single-Round Collaborative Prediction Sets

In this section, we review the key concepts, definitions, and structural results that arise in single-round human–AI collaboration, which will serve as foundational building blocks for our multi-round setup.

Consider a prediction problem with features $X \in \mathcal{X}$ and labels $Y \in \mathcal{Y}$, drawn jointly from an unknown distribution $P$. The goal is to construct a *prediction set* $C(X) \subseteq \mathcal{Y}$ that contains the true label $Y$ with high probability while remaining as small as possible.

In the collaborative setting, a human expert and an AI system jointly construct the prediction set. For a given input $X = x$, the human first proposes a set of plausible outcomes $H(x) \subseteq \mathcal{Y}$, reflecting their proposal. The AI then refines this proposal and outputs a final prediction set

$C(x) := C(x, H(x)) \subseteq \mathcal{Y}$. From the AI's perspective, the human proposal is treated as a black box: the algorithm observes only the set $H(x)$, not how it was generated.

To ensure collaboration is both trustworthy and beneficial, AI's refinement is required to satisfy two guiding principles.

**Counterfactual Harm.** The AI should not degrade correct human judgments. If the human's proposed set already contains the true label, the AI should preserve it with high probability. Formally, for a user-defined $\varepsilon \in (0, 1)$,

$$\mathbb{P}\big(Y \notin C(X) \,\big|\, Y \in H(X)\big) \le \varepsilon. \tag{1}$$

**Complementarity.** The AI should add value precisely when the human misses the correct outcome. If the true label is absent from the human proposal, the AI should recover it with high probability. For a user-defined $\delta \in (0, 1)$,

$$\mathbb{P}\big(Y \in C(X) \,\big|\, Y \notin H(X)\big) \ge 1 - \delta. \tag{2}$$

Among all prediction sets satisfying these requirements, the objective is to minimize expected set size,

$$\min_{C:\mathcal{X}\to 2^{\mathcal{Y}}} \quad \mathbb{E}\big[|C(X)|\big] \tag{3}$$
$$\text{s.t.} \quad \mathbb{P}\big(Y \notin C(X) \,\big|\, Y \in H(X)\big) \le \varepsilon,$$
$$\mathbb{P}\big(Y \in C(X) \,\big|\, Y \notin H(X)\big) \ge 1 - \delta.$$

where $\varepsilon, \delta \in (0, 1)$ control the allowed levels of counterfactual harm and complementarity, respectively.

A key result from (Noorani et al., 2025b) characterizes the optimal solution to this problem in the population setting.

**Theorem 3.1.** *The solution to the optimization problem 3 that minimizes* $\mathbb{E}[|C(X)|]$ *is of the form*

$$C^*(x) = \big\{y \in \mathcal{Y} : s(x, y) \le a^* \mathbf{1}\{y \notin H(x)\}$$
$$+ b^* \mathbf{1}\{y \in H(x)\}\big\}.$$

Given this, several points follow.

(i) The function $s(x, y)$ is a nonconformity score, a standard notion in conformal prediction.

(ii) In the single-round framework of Noorani et al. (2025b), the AI contributes only once at the end of each interaction. Consequently, the AI's behavior within a problem instance does not influence the human's proposal, and offline (exchangeable) calibration techniques remain applicable. This separation breaks down in the multi-round setting: because the AI intervenes repeatedly within each interaction, its prediction sets can influence the human's subsequent behavior. As a result, human behavior evolves in response to the AI's actions, making offline calibration infeasible. For this reason, our analysis is entirely online.

(iii) Despite these differences, we retain a key structural insight from Theorem 3.1. In the single-round setting, the sets take the form $s(x, y) \le a^* \mathbf{1}\{y \notin H(x)\} + b^* \mathbf{1}\{y \in H(x)\}$, where the indicators $\mathbf{1}\{y \notin H(x)\}$ and $\mathbf{1}\{y \in H(x)\}$ arise directly from the definitions in (1) and (2). As we will see in the next section, in the multi-round setting, counterfactual harm and complementarity become user-defined, and these indicators are therefore replaced by more general events defined over the interaction history.

We now proceed to the formal problem formulation for multi-round human–AI collaboration.

# 4. Problem Formulation: Multi-Round Collaboration

In this section, we introduce an interaction protocol and a framework for user-defined requirements in multi-round human–AI collaboration.

## 4.1. Interaction Protocol

We propose a human–AI interaction protocol in an *online* setting, where problems arrive sequentially over time and are handled one at a time.

At each time step $t = 1, 2, \ldots$, a new problem instance $P_t$ arrives, representing a task the human seeks to solve, together with an unknown ground-truth answer $y_t \in \mathcal{Y}$[1]. The correct answer $y_t$ is unknown to both the human and the AI during the interaction.

For each problem $P_t$, the human and the AI engage in a multi-round dialogue. The interaction consists of $N_t \in \mathbb{N}$ rounds, where the stopping time $N_t$ is chosen by the human. At a high level, the human and the AI alternate between proposing candidate answer sets and exchanging textual messages, after which the true answer is eventually revealed.

Formally, for each day $t$ and round $r = 1, \ldots, N_t$:

- the human proposes a pair

$$(H_{t,r}, U_{t,r}),$$

  where $H_{t,r} \subseteq \mathcal{Y}_t$ is a set of candidate answers and $U_{t,r}$ is a textual message;

- the AI responds with a pair

$$(C_{t,r}, A_{t,r}),$$

  where $C_{t,r} \subseteq \mathcal{Y}_t$ is a prediction set refining the human's proposal and $A_{t,r}$ is a textual response.

---

[1]The label space $\mathcal{Y}$ could itself be a function of $t$ in our framework, but for simplicity we avoid introducing that notation.

At the end of day $t$, after the interaction terminates, the ground-truth answer $y_t$ is revealed.

The interaction protocol consists of four components: human textual messages $\{U_{t,r}\}$, AI textual messages $\{A_{t,r}\}$, human prediction sets $\{H_{t,r}\}$, and AI prediction sets $\{C_{t,r}\}$. Among these, our algorithmic focus is exclusively on designing the AI prediction sets. The remaining components are treated as black boxes: we make no assumptions about how human or AI textual behavior evolves, nor about how the human updates their prediction sets in response to the AI. We briefly argue why this modeling choice is both natural and useful.

The textual components $(U_{t,r}, A_{t,r})$ may depend arbitrarily on the full interaction history and may convey any information available to the respective party, such as clarifications, questions, intermediate reasoning, or external context. In particular, the AI's textual responses can be viewed as those of a fixed, pre-trained language model capable of engaging in a dialogue. Treating these textual components as black boxes allows us to decouple language generation from uncertainty quantification and to remain agnostic to how conversational behavior adapts to the interaction.

While the textual components facilitate the flow of information between the two parties, the prediction sets play a distinct role: they represent each party's uncertainty about the ground-truth answer, which is ultimately the most consequential quantity for decision making and downstream action. Through principled communication of uncertainty via prediction sets, both parties can calibrate how much trust to place in each other's inputs, determining whether the collaboration is both harmless and productive.

Accordingly, in the remainder of this paper we focus on how the dynamics between the human sets $\{H_{t,r}\}$ and the AI sets $\{C_{t,r}\}$ should be governed. From an algorithm-design standpoint, we have no direct control over human behavior, and we therefore treat the human prediction sets as black-box objects provided to the system. Therefore, our focus is exclusively on designing the AI's prediction sets.

We now turn to the question of how the AI's prediction sets should be constrained so that they are maximally beneficial to humans in a multi-round setting.

### 4.2. User-Defined Collaboration Requirements

In multi-round interactions, counterfactual harm and complementarity no longer admit a single canonical definition. Even when the high-level goal is clear, preserving correct human judgments and complementing incorrect ones, the precise operational meaning can vary across applications.

Let us first focus on counterfactual harm. To gain intuition, fix a day $t$ and a round $r \geq 2$. Consider the human transcript

of sets up to this point, $H_{t,1:r} := (H_{t,1}, \ldots, H_{t,r})$, and the AI set at round $r$, $C_{t,r} \subseteq \mathcal{Y}_t$.

From the perspective of defining a counterfactual harm requirement at round $r$, the high level goal is to enforce that the AI includes the truth at this round, i.e., that $\mathbf{1}\{y_t \in C_{t,r}\}$ is likely, *conditional on an event indicating that the human "had" the truth in some meaningful sense.* For instance, one might condition on:

- $\mathbf{1}\{y_t \in H_{t,r}\}$, that is human included the truth at round $r$.

- $\mathbf{1}\{y_t \in H_{t,r} \cap H_{t,r-1}\}$, that is human insisted on truth over the last two rounds so far.

- $\mathbf{1}\{\exists\, r' \leq r : y_t \in H_{t,r'}\}$, that is the human proposed the truth at least once.

Many other conditions are possible. Different choices can lead to qualitatively different collaboration dynamics, including different AI set sizes. This motivates a general framework in which counterfactual harm is defined with respect to a user-specified rule, depending on user's preferences in different applications.

**Rule-Based Formulation.** We formalize user-defined counterfactual-harm through a *rule*, a binary function that specifies when the counterfactual-harm constraint is applied. Concretely, a counterfactual-harm rule maps a label $y$ and a complete transcript prefix of human sets for the day, and a round index $r$ to $\{0, 1\}$:

$$R^{\mathrm{CH}}(y, H_{t,1:N_t}, r) \in \{0, 1\},$$

where $H_{t,1:N_t} := (H_{t,1}, \ldots, H_{t,N_t})$ denotes the full transcript prefix of human sets over all $N_t$ round of day $t$, and $r \in \{1, \ldots, N_t\}$ indexes the round under consideration. We interpret $R^{\mathrm{CH}}(y, H_{t,1:N_t}, r) = 1$ as the rule "triggering" at round $r$ of day $t$ if $y$ were the correct label. The examples above correspond to different choices of $R^{\mathrm{CH}}$. Allowing the rule to depend on the full transcript $H_{t,1:N_t}$ allows to capture rules that involve global properties of the interaction, such as whether round $r$ is the terminal round ($r = N_t$), or whether the human ever included a label across the entire dialogue. Since rules are evaluated post-hoc, after the interaction on day $t$ has concluded and the ground truth $y_t$ is revealed, the full transcript is available at evaluation time.

Given a rule, we say that *counterfactual harm occurs on day $t$* if there exists a round in which the rule triggers but the AI fails to include the truth:

$$E_t^{\mathrm{CH}} = \max_{r \in \{1, \ldots, N_t\}} R^{\mathrm{CH}}(y_t, H_{t,1:N_t}, r)\mathbf{1}\{y_t \notin C_{t,r}\}$$

Accordingly, the cumulative number of counterfactual-harm violations up to time $T$ is $\sum_{t=1}^{T} E_t^{\mathrm{CH}}$.

A similar construction applies to complementarity. We formalize user-defined complementarity requirements through

a *verifiable rule*

$$R^{\mathrm{Comp}}\big(y, H_{t,1:N_t}, r\big) \in \{0, 1\},$$

where $R^{\mathrm{Comp}}(y, H_{t,1:N_t}, r) = 1$ indicates that, if $y$ were the correct label, the AI is required to include it at round $r$ in order to complement the human's proposal.

Given a complementarity rule, we say that a *complementarity error occurs on day $t$* if there exists a round in which the rule triggers but the AI fails to include the truth:

$$E_t^{\mathrm{Comp}} = \max_{r \in \{1, \dots, N_t\}} R^{\mathrm{Comp}}(y_t, H_{t,1:N_t}, r)\mathbf{1}\{y_t \notin C_{t,r}\}.$$

The cumulative number of complementarity failures up to time $T$ is then given by $\sum_{t=1}^{T} E_t^{\mathrm{Comp}}$.

The goal of multi-round human–AI collaboration is to design the AI prediction sets $\{C_{t,r}\}$ so as to control counterfactual harm and complementarity errors over time. Concretely, for user-specified tolerances $\varepsilon, \delta \in (0, 1)$, We seek to enforce the following constraints for all horizons $T \geq 1$:

$$\frac{1}{T}\sum_{t=1}^{T} E_t^{\mathrm{CH}} \leq \varepsilon \quad \text{and} \quad \frac{1}{T}\sum_{t=1}^{T} E_t^{\mathrm{Comp}} \leq \delta.$$

## 5. Algorithm and Guarantees

In this section, we present our online algorithm for multi-round human–AI collaboration. Throughout this section, we follow the interaction protocol and asssume two rule functions $R^{\mathrm{CH}}$ and $R^{\mathrm{Comp}}$ are pre-specified by the user, all of which discussed in Section 4.

Recall that the rules $R(y, H_{t,1:N_t}, r)$ depend on the full transcript of human sets, which is only available after the interaction on day $t$ has concluded. However, the AI must construct its prediction sets during the interaction, at each round $r$, before $N_t$ is known. To handle this, at round $r$ we evaluate each rule on the transcript prefix $H_{t,1:r}$, treating the current round as if it were the terminal round:

$$\bar{R}(y, H_{t,1:r}) := R(y, H_{t,1:r}, r).$$

We refer to $\bar{R}$ as the *online activation* of the rule $R$.

**Assumption 5.1.** For each rule $R \in \{R^{\mathrm{CH}}, R^{\mathrm{Comp}}\}$ and its online activation $\bar{R}$, we assume that

$$R(y, H_{t,1:N_t}, r) \leq \bar{R}(y, H_{t,1:r})$$

for all labels $y \in \mathcal{Y}$, all transcripts $H_{t,1:N_t}$, and all rounds $r \leq N_t$.

*Remark* 5.2. Assumption 5.1 is a mild technical condition and is satisfied by a broad class of rules. All rules discussed in Section 4.2 and instantiated in the experiments Section 6 satisfy this assumption.

Let us introduce the notion of a transcript. For each day $t$ and round $r$, let

$$\mathcal{T}_{t,r} := \big(P_t, \{(H_{t,j}, U_{t,j}), (C_{t,j}, A_{t,j})\}_{j=1}^{r}\big)$$

denote the full transcript of the interaction up to round $r$ on day $t$, including the problem instance, human and AI prediction sets, and their textual messages.

We let $s : \mathcal{T} \times \mathcal{Y} \to [0, 1]$ be a nonconformity score that assigns a real-valued score to each label $y$ given a transcript. At day $t$ and round $r$, the AI constructs its prediction set as

$$C_{t,r} = \Big\{ y \in \mathcal{Y}_t : s(\mathcal{T}_{t,r}, y) \leq \tau_t \bar{R}^{\mathrm{CH}}(y, H_{t,1:r})$$
$$+ \lambda_t \bar{R}^{\mathrm{Comp}}(y, H_{t,1:r}) \Big\}.$$

Here, the online activations $\bar{R}^{\mathrm{CH}}$ and $\bar{R}^{\mathrm{Comp}}$ determine which thresholds are active for a given label at round $r$, while the thresholds $\tau_t$ and $\lambda_t$ determine the strictness with which counterfactual harm and complementarity are enforced. Importantly, the thresholds $\tau_t$ and $\lambda_t$ are fixed throughout each day $t$ and therefore indexed only by $t$. That is, within a single problem instance $P_t$, the same thresholds are used across all rounds, while the prediction sets $C_{t,r}$ would still evolve with $r$ as the transcript grows and the score values $s(\mathcal{T}_{t,r}, \cdot)$ change.

We discuss concrete choices of the score function $s(\cdot, \cdot)$ for our applications in Section 6.

Under this parameterization, the problem of designing AI prediction sets reduces to the problem of calibrating the daily thresholds $\{\tau_t, \lambda_t\}$ online, so as to control the cumulative counterfactual harm and complementarity errors.

The thresholds are updated at the end of each day, once the ground-truth label $y_t$ is revealed and the error indicators $E_t^{\mathrm{CH}}$ and $E_t^{\mathrm{Comp}}$ can be evaluated. Specifically, for a step size $\eta > 0$ and target levels $\varepsilon, \delta \in (0, 1)$, the thresholds are updated according to

$$\tau_{t+1} = \max\big\{0, \ \tau_t + \eta\big(E_t^{\mathrm{CH}} - \varepsilon\big)\big\},$$
$$\lambda_{t+1} = \max\big\{0, \ \lambda_t + \eta\big(E_t^{\mathrm{Comp}} - \delta\big)\big\}.$$

The $\max\{\cdot, 0\}$ operator can be interpreted as a projection step that ensures the thresholds remain nonnegative. This projection is needed to handle some degenerate cases in proof of Theorem 5.3, which provides the error control guarantees.

The update rule itself follows a standard approach for online tracking of quantiles. Intuitively, whenever the observed error exceeds its target level, the corresponding threshold

is increased to make future prediction sets more conservative; when the error falls below the target, the threshold is decreased. As the following Theorem shows, this simple update suffices to control cumulative counterfactual harm and complementarity errors in the online setting.

**Theorem 5.3** (Finite-sample online control of collaboration errors). *Under Assumption 5.1 and assuming $\tau_1 = \lambda_1 = 0$ and step size $\eta > 0$, then for every horizon $T \geq 1$,*

$$\frac{1}{T}\sum_{t=1}^{T} E_t^{\mathrm{CH}} \leq \varepsilon + \frac{1+\eta}{\eta T}, \qquad \frac{1}{T}\sum_{t=1}^{T} E_t^{\mathrm{Comp}} \leq \delta + \frac{1+\eta}{\eta T}.$$

Theorem 5.3 shows that our algorithm achieves finite-sample control of both counterfactual harm and complementarity, thereby enforcing user-defined constraints on the collaboration dynamics. Importantly, these guarantees hold in a fully online and distribution-free manner, without any assumptions on the distribution of problems, the human's behavior, or the internal mechanisms of the AI system.

# 6. Experiments

We evaluate our framework in two interactive settings: LLM-simulated agents for scalable verification and human crowd-sourcing for real-world validation. Our objective is twofold. First, we verify that our online algorithm strictly adheres to nominal targets $\varepsilon$ and $\delta$ under non-stationary conditions. We then validate the utility of our organizing principles by showing that counterfactual harm and complementarity are the right metrics to control from the AI side. While we cannot directly dictate human behavior, we find that enforcing these human-centric rules translates to predictable and desirable changes in the human's final decision quality. By varying the strictness of these guarantees, we demonstrate that these two notions are not just mathematical abstractions but are the fundamental levers required to steer human judgment toward better outcomes. Specifically, by algorithmically regulating the AI's uncertainty, we protect humans from abandoning correct intuitions and help them recover correct answers they initially missed.

**Instantiation of Rules and Score Functions.** To instantiate the general framework, each experiment specifies: (i) a pair of verifiable rules $(R^{\mathrm{CH}}, R^{\mathrm{Comp}})$ as in Section 4.2; and (ii) a nonconformity score $s(\mathcal{T}_{t,r}, y) \in [0,1]$ (Section 5). The rules used across both settings are defined as follows: for a label $y \in \mathcal{Y}$ and a transcript history $\mathcal{T}_{t,r}$, the *counterfactual-harm rule* is triggered at any round where the human's latest prediction set already contains the true label:

$$R^{\mathrm{CH}}(y, H_{t,1:N_t}, r) = \mathbf{1}\{y \in H_{t,r}\}.$$

Since this rule depends only on the current round, its online activation coincides with the rule itself: $\bar{R}^{\mathrm{CH}}(y, H_{t,1:r}) =$

$\mathbf{1}\{y \in H_{t,r}\}$. The *complementarity rule* triggers only at the final round $N_t$ if the human's final proposal lacks the label:

$$R^{\mathrm{Comp}}(y, H_{t,1:N_t}, r) = \mathbf{1}\{y \notin H_{t,r}\} \cdot \mathbf{1}\{r = N_t\}$$

These rules ensure that $E_t^{\mathrm{CH}}$ monitors whether the AI fails to include the ground truth the human already considered at any point, while $E_t^{\mathrm{Comp}}$ monitors whether the AI fails to provide the ground truth in the final round when the human is missing it. We adopt this canonical pair of rules throughout the main text, but the framework accommodates a broad family of verifiable rules. For additional rule instantiations, refer to Appendix E.3, which instantiates an alternative pair of counterfactual-harm and complementarity rules.

In both experiments, the AI agents are powered by LLMs that provide probability vectors $p_{t,r}(y) \in [0,1]$ over the label space $\mathcal{Y}_t$ conditioned on the transcript $\mathcal{T}_{t,r}$. To ensure these values constitute a valid probability distribution and to mitigate the impact of LLM hallucinations, we normalize the raw model outputs across the label space. From these normalized probabilities, we derive the standard nonconformity score:

$$s(\mathcal{T}_{t,r}, y) := 1 - p_{t,r}(y).$$

## 6.1. LLM-Simulated Experiments

To evaluate our framework at scale, we first simulate human-AI collaboration using LLMs as proxy agents to model human-like decision making in a controlled environment (Aher et al., 2023); enabling large-scale, reproducible tests of multi-round dialogue dynamics.

**Experimental Design: Medical Diagnosis.** We utilize the DDXPlus dataset (Tchango et al., 2022), which consists of synthetic patient records including demographics, symptoms, and ground-truth diagnoses. Each day $t$ corresponds to one patient case $P_t$ with ground-truth diagnosis $y_t \in \mathcal{Y}_t$.

To ensure collaboration is functionally necessary, we introduce a structural information asymmetry: the *human agent* (GPT-4o-mini) observes demographics and a small set of initial symptoms, while the *AI agent* (DeepSeek-Chat) observes a complementary subset of diagnostic evidence but is blind to demographics. The interaction unfolds through a dynamic dialogue where the human maintains a fixed-size candidate set of two diagnoses. In each round, the human proposes a set $H_{t,r}$ and a textual message $U_{t,r}$, to which the AI responds with a refined prediction set $C_{t,r}$ and its own textual feedback $A_{t,r}$. Following each AI response, the human agent evaluates the new information to decide whether to update their prediction set, request further clarification, or terminate the session. Once the interaction concludes, either by the human's choice or upon reaching a round limit, the human provides a final assessment set. This final set captures the human's ultimate judgment following the di-

alogue, allowing us to evaluate how effectively the AI's uncertainty communication steered the human toward the correct diagnosis.

**Empirical Convergence.** Figure 12 reports the running averages of the realized errors $\{E_t^{\mathrm{CH}}\}$ and $\{E_t^{\mathrm{Comp}}\}$ for a representative target pair. Consistent with Theorem 5.3, the observed averages stabilize near their nominal targets despite the fact that the human agent's dialogue policy, stopping times $N_t$, and set-updates are unconstrained and may vary across trials. Additional settings appear in Appendix E.

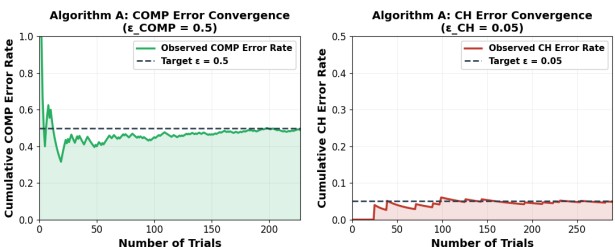

*Figure 1.* Empirical convergence of error rates in the LLM-simulated medical task for complementarity (left) and counterfactual harm (right). Both plots track the cumulative average running error, $\mathrm{AvgError}_t = \frac{1}{t}\sum_{i=1}^{t}\mathbf{1}\{\mathrm{Error}_i\}$, over sequential trials.

**From AI Sets to Human Decisions** Having established that we can control these metrics, we evaluate whether they are meaningful proxies for collaboration quality. By running multiple instances of our algorithm with varying error targets, we observe how the complementarity and counterfactual harm constraints influence the human's final decisions. As illustrated in Figure 2, we observe a direct correlation between the AI's error targets and human outcomes.

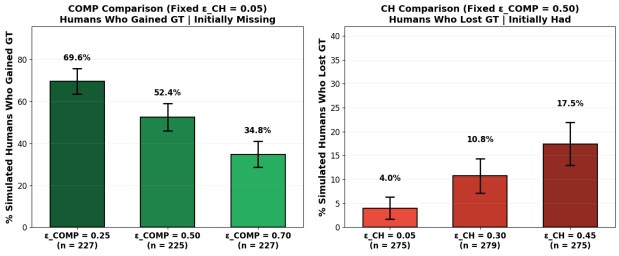

*Figure 2.* Human decision outcomes across varying AI error targets in the LLM-simulated medical task. (Left) Human GT gain rate as a function of the complementarity error target ($\varepsilon_{\mathrm{COMP}}$). (Right) Rate of abandoning a correct initial guess (GT loss) as a function of the counterfactual harm error target ($\varepsilon_{\mathrm{CH}}$).

Specifically, when we increasing the allowed counterfactual harm (CH) results in participants frequently abandoning correct initial guesses. This happens when the AI's set excludes the true label, inducing the human to discard their correct judgment. Conversely, tightening complementarity (COMP) increases the recovery rate—the frequency with

which participants identify the correct label after an initial error. In these cases, the AI's set includes the missed label, which the human then adopts. These results demonstrate that CH and COMP are the fundamental levers of collaboration: by regulating these two metrics in the AI's prediction sets, we achieve predictable improvements in human decisions without the need to model or control human behavior directly.

## 6.2. Crowdsourcing Experiment: Collaborative Shape Counting

While LLM simulations provide a scalable testbed for our algorithm, they cannot fully capture the nuances of human trust, fatigue, or heuristic-driven reasoning. We therefore conduct a human crowdsourcing study to test our framework's real-world robustness. Specifically, we evaluate whether enforcing CH and COMP constraints strictly on the AI's prediction sets reliably improves human decisions and confirming that these principles remain effective when interacting with unpredictable human participants.

**Task Design.** We evaluate our framework using a collaborative visual reasoning task where participants work with an AI assistant to estimate the count of target shapes(e.g. triangles, squares, or stars) within a cluttered image. Each trial begins with the human viewing an image for one second before providing an initial guess as a contiguous range of three integers. This fixed set size is maintained for human's proposals throughout the interaction.

To facilitate collaboration, an AI assistant, powered by a Gemini 2.5 Flash model (Gemini Team, Google, 2024), analyzes a noisy version of the image to produce a distribution over counts, which our algorithm converts into a prediction set for the human. The interaction then enters a second round where the human re-views the image for a 0.5 seconds, observes the AI's set, and revises their guess. The AI then updates its distribution given the conversation history and provides a final set. The interaction concludes with the human submitting a final assessment based on the dialogue. Appendix C provides full interface and parameter details.

We instantiate three AI assistants, each running

|  | $\varepsilon_{\mathrm{CH}}$ | $\varepsilon_{\mathrm{COMP}}$ |
|---|---|---|
| Alg A | 0.05 | 0.50 |
| Alg B | 0.30 | 0.50 |
| Alg C | 0.05 | 0.70 |

*Table 1.* **Algorithm instantiations.** Target rates for counterfactual harm (CH) and complementarity (COMP).

ning our algorithm with different target error rates for counterfactual harm and complementarity: Algorithms A

and B share the same complementarity target but differ in counterfactual harm. Algorithms A and C share the same counterfactual harm target but differ in complementarity.

We ran our study via Prolific[2] involving 1000 total interactions across 50 participants who arrived sequentially. Each participant was randomly assigned to one of three algorithms and completed 20 trials. For each algorithm, we tracked the calibration state globally across all users in a single stream, updating thresholds continuously as new participants joined. This design creates significant non-stationarity due to variations in individual accuracy, speed, and trust, providing a rigorous real-world stress test for our online, distribution-free guarantees.

**Empirical Convergence.** Despite these shifting human dynamics, Figure 3 shows that our algorithm successfully adapted its thresholds to maintain target error rates across the entire population. Additional convergence results for the other instantiations can be found in Appendix E.

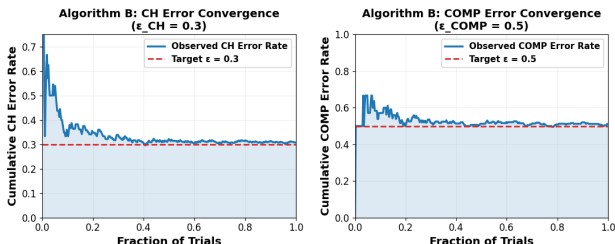

*Figure 3.* Error convergence for Algorithm B in the crowdsourcing study. Both plots track the cumulative average running error, $\text{AvgError}_t = \frac{1}{t}\sum_{i=1}^{t}\mathbf{1}\{\text{Error}_i\}$, where observed rates for counterfactual harm (left) and complementarity (right) stabilize at their nominal values.

**Downstream Results: From AI Errors to Human Decisions** We now evaluate how the algorithm's adherence to counterfactual harm (CH) and complementarity (COMP) targets shapes actual human outcomes. To isolate these effects, participants were required to maintain a fixed set size of three integers for their guesses throughout the trial. This constraint is critical: it ensures that changes in human accuracy are driven specifically by the AI's suggestions rather than fluctuations in the human's own uncertainty or set-size efficiency.

**Theme 1: Direct Impact of AI Errors on Human Behavior.** To determine if CH and COMP serve as effective proxies for collaboration quality, we analyzed trials where these rules were triggered and measured how AI errors altered human outcomes.

We specifically examined whether preventing harm pre-

---
[2]https://www.prolific.co/

serves correct human answers by focusing on trials where the human initially held the ground truth (GT). As shown in Figure 4 (top row), a CH error significantly increases the rate at which humans abandon the correct answer. In Algorithm A, for example, the GT loss rate rises from 5.1% to 35.3% when a CH error occurs. This demonstrates that AI failures to reinforce correct human beliefs can directly induce humans to discard their initial strengths.

We also examined whether AI complementarity directly aids human recovery by focusing on trials where the human initially lacked the ground truth (GT). We partitioned these cases by whether the AI provided the GT or committed a COMP error. The results in Figure 4 (bottom row) show that successful AI complementarity leads to significantly higher recovery rates in humans. In Algorithm B, for instance, humans identified the GT in 35.8% of trials when the AI was complementary, compared to just 6.4% when the AI failed to include it. This gap confirms that providing the missing label in the prediction set is the primary mechanism for helping humans overcome initial errors.

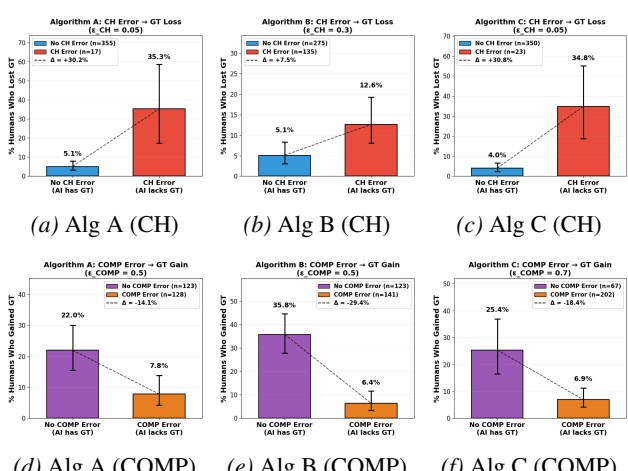

*(a) Alg A (CH)*  *(b) Alg B (CH)*  *(c) Alg C (CH)*

*(d) Alg A (COMP)*  *(e) Alg B (COMP)*  *(f) Alg C (COMP)*

*Figure 4.* Behavioral impact of AI errors in the crowdsourcing task across Algorithms A, B, and C. **Top row:** human GT loss rates compared across trials with and without CH errors. **Bottom row:** human GT gain rates compared across trials with and without COMP errors.

**Theme 2: Comparative Impact of Algorithmic Guarantees.** Beyond the per-error analysis above, consistent with our LLM simulations, we compare different algorithm instances to demonstrate how shifting the counterfactual harm and complementarity error targets globally influence human performance. Comparing Algorithm B's looser counterfactual harm guarantee against Algorithm A's stricter target, we observe a clear correlation: as the AI is permitted to commit more CH errors, participants become significantly more likely to abandon a previously held ground truth (Figure 5a).

A similar trend emerges when comparing the complementar-

ity targets of Algorithms A and C (Figure 5b). We find that a stricter complementarity guarantee directly increases the human recovery rate. Because the human's set size remains fixed at three, these shifts in coverage cannot be attributed to changes in set-size efficiency. Instead, these results confirm that counterfactual harm and complementarity are the primary levers of the interaction. By precisely adjusting the target rates for these two error types, we can predictably steer the final decision quality of the human partner without needing to model their individual behavior or latent trust levels.

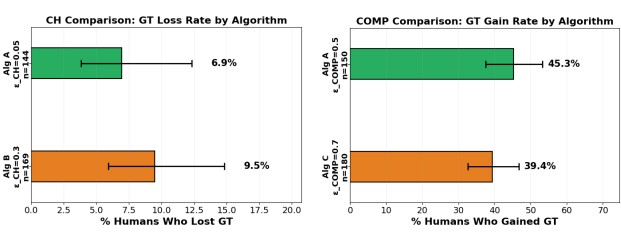

|     (a) CH: Alg A vs Alg B     |     (b) COMP: Alg A vs Alg C     |

*Figure 5.* **Impact of nominal error rates on human outcomes:** (a) Comparison of human's GT loss rates between Algorithm A ($\varepsilon_{\text{CH}} = 0.05$) and Algorithm B ($\varepsilon_{\text{CH}} = 0.30$). (b) Comparison of human's GT gain rates between Algorithm A ($\varepsilon_{\text{COMP}} = 0.50$) and Algorithm C ($\varepsilon_{\text{COMP}} = 0.70$).

## 7. Discussion and Future Work

We introduced a multi-round human–AI interaction protocol, a rule-based framework for specifying user-defined constraints over collaboration dynamics, and an online algorithm with finite-sample guarantees that provably enforces these constraints over a sequence of conversations.

A key limitation is that we control collaboration dynamics by constraining the AI's *prediction sets*, which represent only one mechanism for communicating uncertainty. In settings where outputs are open-ended or do not lie in a well-structured candidate space, other uncertainty representations may be more appropriate. Extending our rule-based control framework and online guarantees to alternative uncertainty objects is a promising direction for future work.

## Acknowledgements

The authors thank EnCORE, the Institute for Emerging CORE Methods in Data Science, for their support. SK additionally acknowledges support from a gift from AWS to Penn Engineering's ASSET Center for Trustworthy AI.

## Impact Statement

This work provides a principled foundation for multi-round human–AI collaboration, introducing user-specified requirements and online guarantees for enforcing them over time. It helps to design interactive AI systems that are provably safer and more helpful in high-uncertainty settings, and offers practitioners a practical way to tune collaboration behavior to reduce harm and improve decision quality.

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

# A. Proofs

*Proof of Theorem 5.3.* We prove the counterfactual-harm bound; the complementarity bound follows by the same argument with the substitutions $(\tau_t, E_t^{\text{CH}}, \varepsilon) \mapsto (\lambda_t, E_t^{\text{Comp}}, \delta)$.

**Step 1: Relating $\tau_{T+1}$ to cumulative counterfactual harm.** By the update rule and the fact that $\max\{0, a\} \geq a$ for all $a \in \mathbb{R}$, we have for every $t$,

$$\tau_{t+1} = \max\{0, \ \tau_t + \eta(E_t^{\text{CH}} - \varepsilon)\} \ \geq \ \tau_t + \eta(E_t^{\text{CH}} - \varepsilon).$$

Summing over $t = 1, \ldots, T$ and telescoping yields

$$\tau_{T+1} - \tau_1 \ \geq \ \eta \sum_{t=1}^{T} (E_t^{\text{CH}} - \varepsilon),$$

or equivalently,

$$\sum_{t=1}^{T} E_t^{\text{CH}} \ \leq \ \varepsilon T + \frac{\tau_{T+1} - \tau_1}{\eta}. \tag{4}$$

**Step 2: Uniform upper bound on $\tau_t$.** We claim that $\tau_t \leq 1 + \eta$ for all $t \geq 1$ when $\tau_1 = 0$. If $\tau_t \geq 1$, then $E_t^{\text{CH}} = 0$. Indeed, whenever $R^{\text{CH}}(y_t, H_{t,1:r}) = 1$, we have

$$\tau_t R^{\text{CH}}(y_t, H_{t,1:r}) + \lambda_t R^{\text{Comp}}(y_t, H_{t,1:r}) \ \geq \ \tau_t \ \geq \ 1,$$

since $\lambda_t \geq 0$ by construction. Because $s(\mathcal{T}_{t,r}, y_t) \in [0, 1]$, this implies $y_t \in C_{t,r}$ at every round where the rule triggers, hence

$$E_t^{\text{CH}} = \max_{r \in \{1, \ldots, N_t\}} R^{\text{CH}}(y_t, H_{t,1:r}) \, \mathbf{1}\{y_t \notin C_{t,r}\} = 0.$$

Therefore, if $\tau_t \geq 1$ then the update satisfies

$$\tau_{t+1} = \max\{0, \tau_t - \eta\varepsilon\} \leq \tau_t,$$

so $\tau$ cannot increase while above 1.

If instead $\tau_t < 1$, then $E_t^{\text{CH}} \leq 1$ and thus

$$\tau_{t+1} = \max\{0, \ \tau_t + \eta(E_t^{\text{CH}} - \varepsilon)\} \leq \tau_t + \eta < 1 + \eta.$$

Combining the two cases yields $\tau_t \leq 1 + \eta$ for all $t$, and in particular $\tau_{T+1} \leq 1 + \eta$.

**Step 3: Concluding the finite-sample bound.** Plugging $\tau_1 = 0$ and $\tau_{T+1} \leq 1 + \eta$ into (4) gives

$$\sum_{t=1}^{T} E_t^{\text{CH}} \ \leq \ \varepsilon T + \frac{1 + \eta}{\eta}.$$

Dividing by $T$ proves

$$\frac{1}{T} \sum_{t=1}^{T} E_t^{\text{CH}} \leq \varepsilon + \frac{1 + \eta}{\eta T}.$$

The complementarity bound follows identically. $\qquad\square$

# B. Extended Related Works

**Conformal Prediction**   Modern conformal prediction (Vovk et al., 1999; 2005; Saunders et al., 1999a;b) builds on the classical work on tolerance regions in statistics (Wilks, 1941; Scheffe & Tukey, 1945) to provide distribution free finite sample validity for prediction sets. Over the past two decades CP has become a standard tool in machine learning in classification, regression, and recently large language models (Papadopoulos et al., 2002; Romano et al., 2019; Lei et al., 2017; Romano et al., 2020; Angelopoulos et al., 2022; 2025; Noorani et al., 2025a; Quach et al., 2024). There is a growing body of work with the aim to improve different aspects of conformal prediction, including length efficiency of the prediction sets (Kiyani et al., 2024; Stutz et al., 2022; Noorani et al., 2025c; Sadinle et al., 2019), online guarantees (Gibbs & Candès, 2021; Gibbs & Candès, 2024; Angelopoulos et al., 2024; Bhatnagar et al., 2023; Ramalingam et al., 2025), and extension to decision-theoretic frameworks (Lekeufack et al., 2024; Kiyani et al., 2025; Lindemann et al., 2025; Cortes-Gomez et al., 2025). Particularly, the decision-theoretic angle supports the hypothesis that prediction sets, specifically with guarantees accompanied by conformal prediction, are a useful tool for human-AI collaboration in high stakes applications, where reliable uncertainty estimates are essential for enabling trust and communication.

**Prediction sets for Human-AI Decision Support**   A recent growing body of work, considers prediction sets as a structured interface for collaboration and human decision making support. (Straitouri et al., 2023) formalizes how conformal prediction sets can improve expert prediction in multiclass classification. (Benz & Rodriguez, 2025) propose human aligned calibration over standard calibration which they argue ensures that ai generated prediction sets are calibrated specifically on instances where human is likely to be incorrect. (Babbar et al., 2022) empirically evaluate sets as a form of advice in human-AI sets, and show that sets can improve accuracy compared to single label predictions. (Hullman et al., 2025) analyze CP from a decision theoretic perspective and examine tensions between coverage guarantees and decision relevant uncertainty. (Straitouri et al., 2024) analyze prediction sets based decision support through the lens of counterfactual harm, and formalize the concerns that requiring humans to select from machine provided sets may degrade performance. (Toni et al., 2024) propose a greedy algorithm to construct prediction sets that empirically improved human accuracy compared to standard conformal prediction sets. (Wang et al., 2022) study calibrated subset selection, and propose a distribution-free procedure that improves upon standard calibration by producing near-optimal shortlists with a target expected number of qualified candidates. These works study how humans use AI-generated sets to reach final judgments—the human is the end decision-maker receiving a single set. Our framework, following (Noorani et al., 2025b), incorporated the human input and treats the prediction set itself as the collaborative output where the human proposes a set, and the AI refines it. Moreover, these works analyze single interactions per instance, leaving open how to handle iterative refinement across multiple rounds where the human may update beliefs, ask clarifying questions, or revise proposals.

**Agreement Protocols**   Agreement protocols formalize multi-round collaboration as iterative belief exchange until consensus. The foundational result is Aumann's agreement theorem (Aumann, 1976), which states that two perfectly informed Bayesians with a common prior, different observations, and common knowledge of each other's posteriors must share the same posterior. Subsequent work has given finite-time convergence protocols through which the different parties engage in a conversation about their beliefs about the outcome (Geanakoplos & Polemarchakis, 1982; Aaronson, 2004). Recent work (Collina et al., 2025) generalize prior results and introduce tractable agreement protocols that replace Bayesian rational, common assumption in earlier formulations, with statistically efficient calibration condition that enables agreement theorems without distributional assumptions. (Collina et al.) extend upon these protocols and achieve information aggregation as well as in the setting where the two agents have asymmetric information.

Our framework differs from agreement protocols in both interface and objective. First, these protocols require both agents to maintain and communicate probabilistic beliefs or expected values; our setting requires only that the human provide a prediction set, with no probabilities attached. Second, agreement protocols focus on belief convergence and information aggregation—the goal is for agents to reach consensus. We prioritize different objectives: ensuring the AI's output does not harm correct human judgments (counterfactual harm) while complementing incorrect ones (complementarity). These objectives do not reduce to reaching consensus and require fundamentally different algorithmic machinery.

**Learning to Defer**   Another related area is "Learning to Defer (L2D)", where an algorithmic tool learns when to defer a final decision to an expert (human). Early work by (Madras et al., 2018) framed this as a mixture-of-experts problem, jointly training a classifier with a deferral mechanism. (Mozannar & Sontag, 2021) extended this with a decision-theoretic formulation, training models to complement human strengths rather than maximize accuracy alone. (Verma & Nalisnick, 2022) showed that standard training objectives can fail to produce optimal deferral policies and proposed consistent

surrogate losses guaranteeing Bayes-optimal deferral. (Raghu et al., 2019) frame automation as an instance-level triage problem and optimally assigning cases to humans vs. algorithms based on estimates of their respective errors. Extensions address various settings: (Hemmer et al., 2023; Keswani et al., 2021) study deferral with multiple experts, while (Wilder et al., 2021) emphasize that humans and models are not independent and introduce dependent Bayes optimality to exploit correlations. (Okati et al., 2021) formulate differentiable learning under triage with exact optimality guarantees. (Charusaie et al., 2022) propose training-free deferral using conformal prediction to allocate decisions among experts. Most recently, (Arnaiz-Rodriguez et al., 2025) introduce collaborative matching for selective deferral.

The L2D literature primarily asks who should decide on a given instance: the model or the human. These methods improve performance through selective abstention, which is fundamentally different from our objective. We do not focus on task allocation; instead, we focus on joint decision-making. From a human-centric perspective, our work addresses how the AI should incorporate human feedback to build a prediction set that reflects the strengths of both parties. While L2D determines whether a collaboration should occur, our framework governs how that collaboration proceeds once it has begun unfolding over multiple rounds.

**Theoretical Frameworks for Human-AI Complementarity**   The concept of human-AI complementarity—where a joint system surpasses the capabilities of either agent in isolation—has been a central focus of recent literature. (Bansal et al., 2021) provided a formal grounding for this synergy, investigating the specific team dynamics that enable AI to add value beyond standalone accuracy. Other researchers have approached this through a Bayesian lens, identifying the criteria under which collective belief updating leads to superior outcomes (Steyvers et al., 2022). This has led to broad taxonomies that categorize how human cognitive intuition and machine learning precision can be optimally fused across different collaborative tasks (Rastogi et al., 2023). Despite these theoretical frameworks, empirical results suggest that achieving such synergy is non-trivial. (Vaccaro et al., 2024) highlights a persistent gap where human-AI dyads often fail to outperform the more capable individual agent, even though they typically exceed baseline human performance.

A second research thread examines how algorithms actively shape human choices through ranking and recommendations. While (Cowgill & Stevenson, 2020) analyze the strategic impact of recommendations on decision-makers, others have utilized the Mallows model to study the interplay between algorithmic outputs and human preferences (Donahue et al., 2024; Kleinberg & Raghavan, 2021).

More recently, human-AI collaboration has been framed as a sequential learning problem. (Chan et al., 2019) introduced the "assistive multi-armed bandit," where a robot learns to aid a human agent who is themselves discovering task rewards. Similar "two-player" perspectives are explored by (Bordt & Von Luxburg, 2022), who emphasize the challenges of private information and opacity in shared decision tasks. In recommendation systems, work by (Agarwal & Brown, 2022) and (Yao et al., 2023) explores how menu selection and creator competition influence users with adaptive preferences or random utility models. Furthermore, (Tian et al., 2023) and (Gao et al., 2025) represent a shift toward modeling the internal dynamics of human learning and human-centered collaboration, treating the human as a teammate whose mental state evolves over time.

Our works departs from the above paradigms in three fundamental ways: (i) Existing models typically rely on parametric assumptions—such as Bayesian updates, Mallows noise, or specific internal mental models. In contrast, we treat the human as a complete black box. We make no assumptions about the human's rationality, learning speed, or internal state. (ii) our guarantees are distribution-free. Using online calibration, we ensure that our collaboration constraints remain valid even under arbitrary, non-stationary shifts in human behavior or problem distributions. (iii) Lastly, different from prior works that focus on reconciling symmetric agents, we adopt an asymmetric, human-centric view. Our framework defines complementarity and "counterfactual harm" through set-based outcomes: the AI's goal is to recover what the human misses without degrading what the human already knows. This formulation allows us to provide rigorous, finite-sample guarantees on the reliability of the interaction.

A more distantly related line of work focuses on incentivized exploration. Here, the AI acts as an information designer, strategically revealing or hiding data to steer humans toward better actions. This includes limiting the AI's exploration so users don't get frustrated and quit the platform (Bastani et al., 2021), or providing incentives to convince short-sighted humans to make fairer, less biased decisions (Kannan et al., 2017). Other researchers explore how to selectively reveal information to encourage humans to try new options (Immorlica et al., 2024; Hu et al., 2022). While these studies provide valuable insights into human-AI dynamics, our approach is built on a different set of priorities. Instead of managing a population of users by strategically revealing information to new people at each step, we focus on an ongoing dialogue over multiple rounds. Our goal isn't to manipulate what a person sees or direct them toward a choice, but rather to construct a

reliable, evolving interaction.

**Human Reliance on Algorithmic Predictions**    A substantial literature studies how humans actually use algorithmic predictions, and understanding how the timing and presentation of AI outputs influence human judgment. (De-Arteaga et al., 2020) empirically show that human practitioners can often identify and override erroneous algorithmic scores in high-stakes settings, emphasizing the critical importance of maintaining human autonomy. In a similar vein, (Benz & Rodriguez, 2025) examine how the presentation of confidence metrics impacts a user's inclination to depend on AI predictions, finding that the perceived alignment between the human and the machine influences the utility of the tool. (McLaughlin & Spiess, 2025) argue that algorithmic suggestions often function as behavioral defaults, where the perceived cost of deviating from a recommendation fundamentally shifts human preferences. This anchoring effect is further detailed by (Fogliato et al., 2022), who found that the specific sequence of interaction—whether AI results are displayed before or after a provisional human judgment—significantly alters diagnostic agreement and perceived utility in clinical radiology. To mitigate such overreliance, (Vasconcelos et al., 2023) propose a cost-benefit framework illustrating that explanations can curb anchoring if they sufficiently lower the cognitive effort required for a human to verify the AI's logic.

(Chen et al., 2023) categorize how internal heuristics about task outcomes and machine limitations determine whether a user successfully applies an explanation to override a false prediction. Beyond the content of the AI's output, the structural protocol of the interaction itself plays a critical role, as evidenced by the study of double-reading workflows by (Cabitza et al., 2021). In the domain of software engineering, (Mozannar et al., 2024) utilize a utility-theoretic approach to selectively withhold code suggestions that have a high likelihood of rejection, thereby reducing the verification burden on the programmer and improving overall efficiency.

This empirical body of work is complementary to ours. While these studies investigate how humans actually behave when using AI, we focus on whether counterfactual harm and complementarity are the correct objectives to regulate in multi-round interactions. We design an algorithm to strictly enforce these objectives without making any assumptions about the human's internal logic, and show that by controlling these AI-side metrics, we can indirectly guide the collaboration toward better outcomes, even though we treat the human as a black box and exert no direct control over their choices.

## C. Crowdsourcing Study Details and Interface

This section provides additional details on the Collaborative Shape Counting crowdsourcing experiment described in Section 6.2, including the recruitment procedure, task interface, and image generation methodology.

### C.1. Participant Recruitment and Study Protocol

We conducted our crowdsourcing study through Prolific[3], an online platform for recruiting research participants. A total of 50 participants were recruited, with participants assigned to each of the three algorithm conditions (A, B, and C) randomly. Participants were required to be fluent in English and have a minimum approval rate of 95% on prior Prolific studies.

**Sequential Participation Protocol.**    A critical aspect of our experimental design is that participants were required to complete the study *sequentially*. We configured the Prolific study to allow only one active participant at any given time. This constraint is essential for maintaining the validity of our online threshold updates: because our algorithm adapts its confidence thresholds based on cumulative error rates across all prior interactions, concurrent participation would introduce race conditions and potentially violate the sequential guarantees of our framework. Once a participant completed all assigned 20 trials, the next participant in the queue was permitted to begin.

**Algorithm Assignment.**    Upon starting the study, each participant was randomly assigned to one of three algorithm configurations using a round-robin assignment scheme tracked via a Redis database.[4] This ensured balanced allocation across conditions while maintaining the sequential integrity of each algorithm's threshold state. The assignment was performed server-side, and participants were unaware of which algorithm variant they were interacting with. Each algorithm instance maintained its own global state, meaning all participants assigned to Algorithm A contributed to a single, continuous stream of interactions with shared adaptive thresholds.

---

[3]https://www.prolific.com/
[4]https://redis.io

**Compensation.** Participants were compensated at a rate of $12 USD per hour, with the median completion time of approximately 15 minutes for the full 20-trial session.

**Task Description.** Figure 6 shows the task instructions presented to participants at the beginning of the study. The instructions explain the collaborative nature of the task, the structure of each trial (two rounds of interaction with the AI assistant), and the mechanics of submitting guess ranges.

### C.2. Image Generation and Difficulty Levels

We generated a dataset of synthetic images containing three types of shapes: triangles, squares, and stars. Each image was procedurally generated with varying numbers of shapes distributed across the canvas with randomized positions, sizes, and colors. The use of synthetic images allowed precise control over ground-truth counts and enabled systematic manipulation of task difficulty. Images were organized along two dimensions: *bucket* (based on the total number of shapes) and *difficulty* (based on visual complexity). Specifically:

- **Bucket:** Images were divided into three buckets based on the number of distinct shape types present in the image:

    - Bucket 1: Images containing only one shape type (e.g., only triangles)
    - Bucket 2: Images containing two distinct shape types (e.g., triangles and squares)
    - Bucket 3: Images containing all three shape types (triangles, squares, and stars)

- **Difficulty:** Within each bucket, images were further classified into three difficulty levels based on the total number of shapes present:

    - Easy: Low total shape count (fewer objects to track)
    - Medium: Moderate total shape count
    - Hard: High total shape count (many objects competing for attention)

Figure 7 presents representative examples from each bucket-difficulty combination, illustrating the range of visual complexity in our dataset.

To create information asymmetry between the human and AI participants, the AI assistant, powered by Gemini 2.0 Flash-Lite (Gemini Team, Google, 2024), received a degraded version of each image. Specifically, we applied a "salt-and-pepper" noise filter at an extreme intensity level, which randomly replaces a substantial fraction of pixels with either black or white values. This degradation ensures that the AI's probability estimates are imperfect, creating a realistic scenario where neither the human nor the AI has complete information. Figure 8 shows examples of the noisy images provided to the AI alongside their original counterparts.

### C.3. User Interface

The study was conducted through a custom web application. Below we describe the key interface states encountered during each trial. After the initial image viewing period (1 second in round 1, 0.5 seconds in round 2), the image is hidden and participants are prompted to enter their guess. Figure 9 shows this intermediate state, where participants can see the AI's prediction set from the current round and must submit their own three-integer range estimate. The interface displays the current round number, the target shape type, and the AI's suggested range. Upon submitting their final guess, participants are shown a summary of the complete interaction. Figure 10 displays this completion state, which reveals: (1) the original image for reference, (2) the full history of both rounds including the human's guesses and the AI's prediction sets, (3) the ground-truth count, and (4) whether the participant's final guess contained the correct answer. This feedback helps participants calibrate their performance across trials.

To ensure data quality, we implemented several safeguards: (i) **Attention checks:** Participants who submitted identical guesses across more than 80% of trials were flagged for review. (ii) **Response time filtering:** Trials with response times below 500ms (indicating random clicking) were excluded from analysis. Lastly (iii) **Completion requirement:** Only participants who completed all 20 trials were included in the final analysis.

# D. LLM-Simulated Experiment Details

In this section we report the exact prompt templates used to the the LLM-simulated medical diagnosis task for reproducibility. Our setup instantiates two agents. (i) a *Human* agent (GPT-4o-mini): The Human agent only observes patient demographics and chief complaint / initial symptoms. (ii) an *AI* agent (DeepSeek-Chat). Importantly, the AI does not observe the patient demographic information. Both agents output **JSON only**.

**Prompt Templates**    As for the notation clarity, we use the following placeholders inside templates:

- {{LABEL_SPACE}}: the full diagnosis label list (size $|\mathcal{Y}|$).

- {{HUMAN_INFO}}: demographics + chief complaint/initial symptoms (what the Human can see).

- {{AI_INFO}}: full clinical picture (what only the AI can see).

- {{ROUND}}: current round number.

- {{HISTORY}}: truncated conversation history (most recent turns).

- {{HUMAN_SET}}: Human's current differential set (from their top-2).

- {{AI_SET}}: AI's current prediction set.

---

**AI Agent (DeepSeek-Chat) — System Prompt**

```
You are an AI clinical decision support system helping with differential diagnosis.

Possible Diagnoses (|𝒴| conditions)
{{LABEL_SPACE}}

Your Information
You have access to the some clinical evidences including symptoms, physical exam
findings, and test results.
The human (physician) can see the patient's age, sex, and chief complaint with
initial symptoms.

Return Format (JSON only)
{
  "reasoning": "<1--2 sentences about your clinical reasoning>",
  "probabilities": [{"diagnosis": "<diagnosis>", "prob": <0--1>}, ...],
}

IMPORTANT:
- The "diagnosis" field MUST be one of the exact names from the list above.
- Include ALL diagnoses with meaningful probability (> 1%).
- The probabilities SHOULD sum to approximately 1.0.
- Use clinical evidence to narrow the differential.
```

---

**Human Agent (GPT-4o-mini) — System Prompt (Round 1: Initial Assessment)**

You are a physician seeing a patient for the first time.

**Possible Diagnoses ($|\mathcal{Y}|$ conditions)**
{{LABEL_SPACE}}

**Your Information**
You can ONLY see the patient's demographics and chief complaint/initial symptoms.
You do NOT have access to detailed clinical findings, exam results, or tests.

**Your Task (Round 1 --- Initial Assessment)**
Based ONLY on the initial presentation, provide your top 2 most likely diagnoses.

**Return JSON:**
{
  "reasoning":  "<Your clinical reasoning based on demographics and chief
complaint>",
  "top_2_diagnoses":  ["<most_likely>", "<second_likely>"]
}

**RULES:**
1.  top_2_diagnoses MUST contain EXACTLY 2 items
2.  Each item MUST be a diagnosis name COPIED EXACTLY from the list above
3.  Use your clinical judgment based on the information available

---

**Human Agent (GPT-4o-mini) — System Prompt (Rounds $t \geq 2$: Collaborative Updates)**

```
You are a physician collaborating with an AI clinical decision support system.

Possible Diagnoses (|𝒴| conditions)
{{LABEL_SPACE}}

Your Information
- You can see the patient's demographics and chief complaint
- The AI has access to detailed clinical findings you cannot see

COLLABORATION RULES:

1.  THINK ABOUT THE AI'S CLINICAL EVIDENCE
- The AI has physical exam findings, test results, and detailed symptoms
- If the AI suggests a diagnosis you didn't consider, the AI likely has clinical
evidence suggesting the diagnosis.
- The AI does not have access to the patient demographics, his judgments are based on
complementary evidence you cannot observe.

2.  UPDATE YOUR ASSESSMENT EACH ROUND
- If the AI suggests diagnoses you didn't consider, add them to your set if you
believe the diagnosis is likely.
- DO NOT stubbornly repeat your previous answer.  Reevaluate each round based on
whether the AI is disagreeing or agreeing with your proposal.

3.  STOPPING RULES --- ONLY stop (should_continue=false) if:
- Your LIKELY diagnosis appears in the AI's set ( You agree with AI ) AND
- The AI's set is reasonably small given the inherent uncertainty in the case.

4.  MUST CONTINUE (should_continue=true) if:
- Your you DO NOT believe the AI's set contains the true diagnosis.
- The AI's set is too large to be helpful for a clinical assessment.

Return JSON:
{
  "reasoning":  "<How does the AI's suggestion change your thinking?>",
  "top_2_diagnoses":  ["<most_likely>", "<second_likely>"],
  "confidence":  <0.0--1.0>,
  "ai_set_analysis":  "<What clinical findings might the AI have seen?>",
  "should_continue":  true or false,
  "why_stop_or_continue":  "<If confident the AI set contains the true diagnosis,
stop; if not confident, continue and explain what is missing>"
}

CRITICAL RULES:
- top_2_diagnoses MUST contain exactly 2 DIAGNOSIS NAMES from the list
- You MUST reassess your top_2_diagnoses each round based on the conversation,
available information, and your belief.
```

**User-Message Context Templates** In addition to system prompts, each agent receives a structured *user message* that (i) provides the visible patient context and (ii) carries forward the collaboration state (sets and limited recent history). We include these templates for completeness.

---

## Human Agent — User Message (Round 1 Context)

**Patient Presentation**

{{HUMAN_INFO}}

**Round 1 --- Initial Assessment**
Based ONLY on the information above (demographics and chief complaint), provide your
top 2 most likely diagnoses.

**Note:** You do NOT have access to detailed clinical findings, physical exam results,
or additional symptoms.

---

## AI Agent — User Message (Round {{ROUND}} Context)

**Patient Information**

**The Physician has access to demographics information you cannot see!**
{{HUMAN_INFO}}

**Clinical Evidence Available (Only YOU can see this):**
{{AI_INFO}}

**Round {{ROUND}}**
Physician's current differential: {{{HUMAN_SET}}}

Use the clinical evidence available to provide an accurate differential diagnosis.
The physician can only see demographics and the chief complaint --- you can see
complementary clinical evidence.

**Conversation History (most recent turns):**
{{HISTORY}}

---

## Human Agent — User Message (Round {{ROUND}} Context)

**Patient Information (What YOU can see)**

{{HUMAN_INFO}}

**Round {{ROUND}} --- AI's Differential (|AI_SET| diagnoses):**
The AI has access to the COMPLEMENTARY clinical evidence (all symptoms, physical exam
findings, etc.) that you cannot see.
Based on that clinical evidence, the AI suggests these diagnoses:
{{AI_SET}}

**Recent History (most recent turns):**
{{HISTORY}}

**IMPORTANT:**
1. The AI has clinical findings you don't have --- their suggestions are INFORMED by
complementary evidence
2. If the AI suggests a diagnosis you didn't expect, they likely believe they have
supporting symptoms/signs

**STOPPING CHECK:**
- Do you believe the AI's set contains the TRUE diagnosis?
- If YES and AI set size is reasonably small: You may stop (should_continue=false)
- If NO: You MUST continue (should_continue=true)

**Medical Task Context: DDXPlus Case Construction and Interaction Protocol:** We build each episode of interaction from a single DDXPlus patient record, mapping the case to a fixed diagnosis label space $\mathcal{Y}$ and enforcing the information asymmetry described above. Each episode proceeds according to the following multi-round protocol:

1. **Round 1 (Human initial):** The human agent receives demographics and initial symptoms and outputs an initial top-2 diagnosis set.

2. **Round $t$ (AI response):** The AI assistant outputs a full probability distribution over all diagnoses. To mitigate potential LLM hallucinations and ensure a valid distribution, we post-process and normalize these probabilities such that they sum to 1. The conformal algorithm then utilizes these normalized probabilities to construct a prediction set $\mathcal{C}_t$ based on the current thresholds for counterfactual harm and complementarity.

3. **Round $t+1$ (Human update):** The human agent reviews the AI's prediction set, updates their top-2 differential, and decides whether to terminate the session based on the pre-defined stopping rules.

4. **Termination and Update:** The interaction continues until the human agent decides to stop or the interaction reaches a maximum of $MAX\_ROUNDS = 6$. Once the session concludes, the ground truth (GT) for the case is revealed. The online adaptive algorithm then observes whether a CH or COMP error occurred during the session and updates the corresponding thresholds to maintain the target error rates for the next episode.

## E. Extended Experimental Results

### E.1. Extended Crowdsourcing Convergence Results:

We present the empirical convergence of the adaptive thresholds for Algorithms A and C in the Collaborative Shape Counting task. As demonstrated in Figure 11, the online algorithm successfully regulates the target error rates across different configurations. The stable convergence of the cumulative average running error to the pre-specified $\varepsilon$ targets confirms the robustness of the adaptive mechanism in real-world human-AI interactions.

### E.2. Extended LLM Simulation Convergence Results:

For further validate the finite-sample guarantees in the medical diagnosis task, we provide convergence results for additional configurations of the target error rates.

### E.3. Alternative Rule Instantiations

The framework of Section 4.2 is agnostic to the specific verifiable rules used to define counterfactual harm (CH) and complementarity (Comp). To illustrate this flexibility, we instantiate a second pair of rules and show that (i) our procedure attains the prescribed CH and Comp error targets under both, and (ii) the choice of rule induces distinct but interpretable effects on the human–AI interaction. The nonconformity score is unchanged throughout, $s(\mathcal{T}_{t,r}, y) = 1 - p_{t,r}(y)$.

**Alternative rule definitions** The alternative *counterfactual-harm rule* triggers only when the human retains the true label across two consecutive rounds, rather than at every round in which the human holds it:

$$R^{\mathrm{CH}}(y, H_{t,1:N_t}, r) = \mathbf{1}\{y \in H_{t,r}\} \cdot \mathbf{1}\{y \in H_{t,r-1}\}, \qquad r \geq 2.$$

Since both indicators are determined by round $r$, the online activation again coincides with the rule itself. The alternative *complementarity rule* moves the trigger from the human's final proposal to their first one:

$$R^{\mathrm{Comp}}(y, H_{t,1:N_t}, r) = \mathbf{1}\{y \notin H_{t,r}\} \cdot \mathbf{1}\{r = 1\}.$$

Figure 13 reports convergence plots under both alternative rules. In each case the empirical CH and Comp error rates track their targets, confirming that the guarantees of Section 5 hold beyond the specific instantiation used in the main text.

**Effect on human behavior.** The two rules also produce intuitive differences in how errors translate into outcomes. For CH, the alternative rule activates only when the human commits to the true label for two consecutive rounds. A given CH error rate is therefore less costly under this rule: the AI is permitted to drop the ground truth only on cases where the human

is firmly committed and thus unlikely to be swayed by the AI's set. Under the original rule the same error rate applies to weakly committed cases, where the human is more likely to follow the AI and abandon the ground truth. Consistent with this, a comparatively high CH error of $45\%$ has limited downstream impact under the alternative rule.

For complementarity, the original rule controls the human's exposure to the ground truth at their final round, whereas the alternative rule controls it at their first. We observe a smaller gain in ground-truth recovery under the alternative rule, which matches intuition: supplying the missing label in the last AI-provided set acts directly on the human's final decision, while supplying it in the first round leaves the remainder of the conversation free to dilute the effect. Summary figures for both effects are provided in Figure 14.

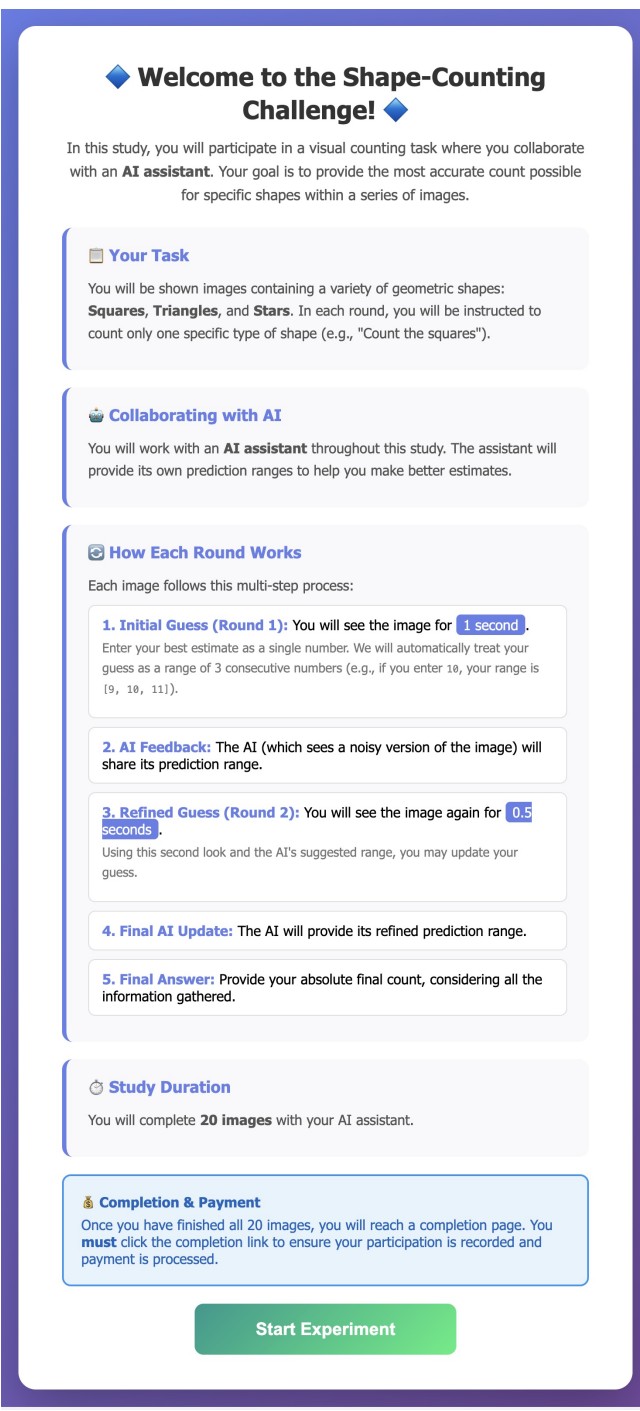

*Figure 6.* **Task instructions presented to participants at the start of the crowdsourcing study.** The instructions describe the Collaborative Shape Counting Task, explain the two-round interaction structure, and clarify how to interpret and respond to the AI assistant's predictions.

*Figure 7.* Sample images from the shape counting dataset, organized by bucket (rows) and difficulty (columns). Bucket 1 contains images with low shape counts (3–15), Bucket 2 contains medium counts (16–30), and Bucket 3 contains high counts (31–50). Difficulty increases from left to right, with "Hard" images featuring greater shape density and visual overlap.

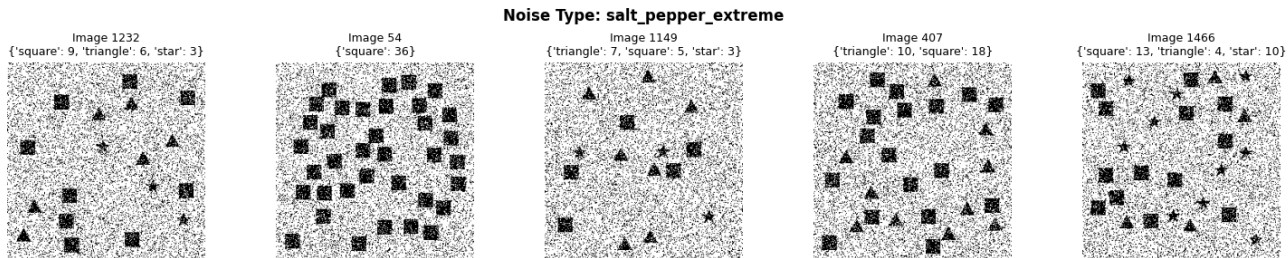

*Figure 8.* Sample visualization of noise-degraded images versions (bottom row) used as input for the AI assistant. The extreme salt-and-pepper noise obscures fine details, ensuring the AI's predictions are uncertain and creating a meaningful collaboration dynamic where both human and AI contributions are valuable.

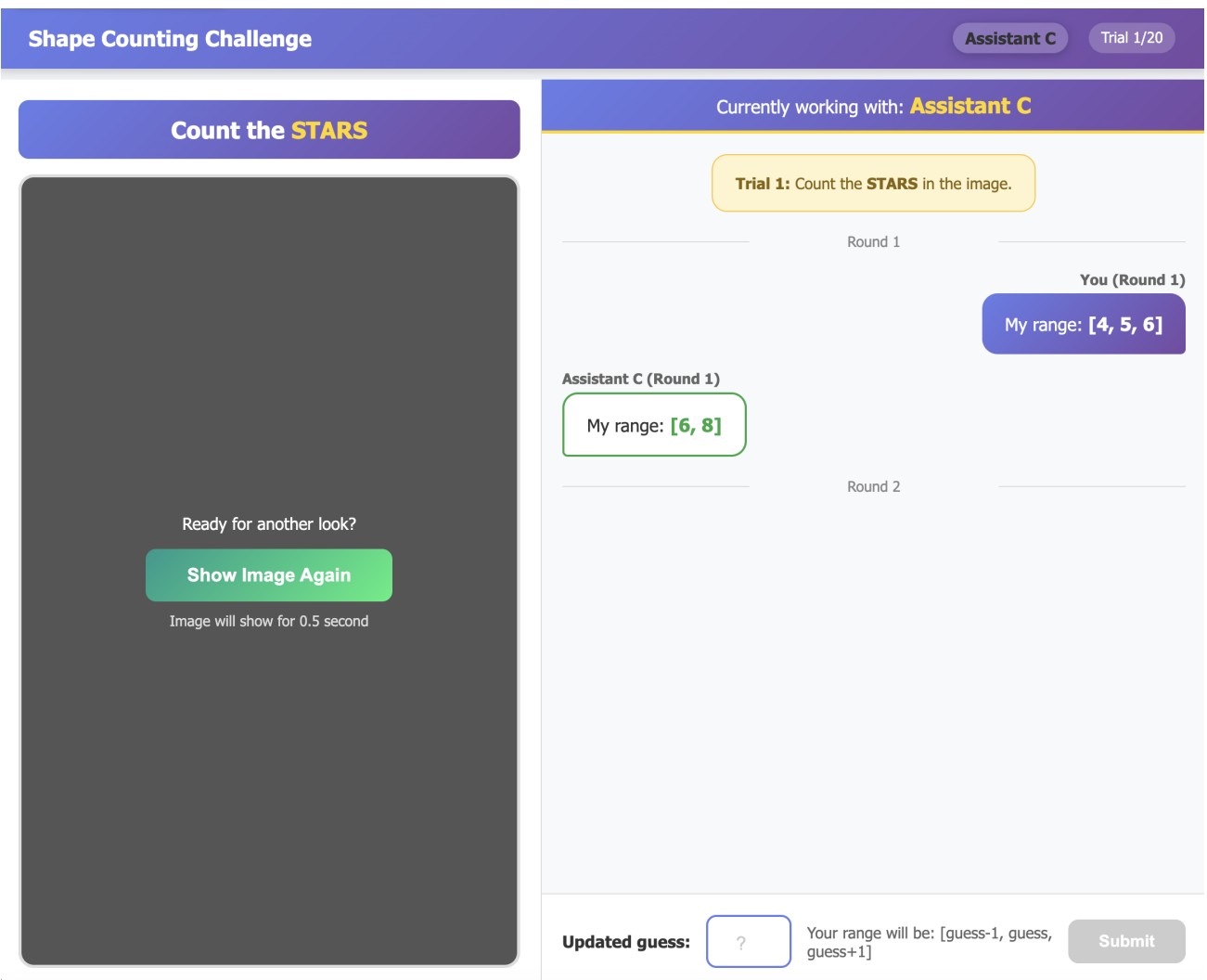

*Figure 9.* Intermediate interaction state of the user interface. The image is no longer visible, and the participant must submit a guess (a contiguous range of three integers) after observing the AI assistant's prediction set. The interface shows the target shape type, current round, and the AI's suggested range.

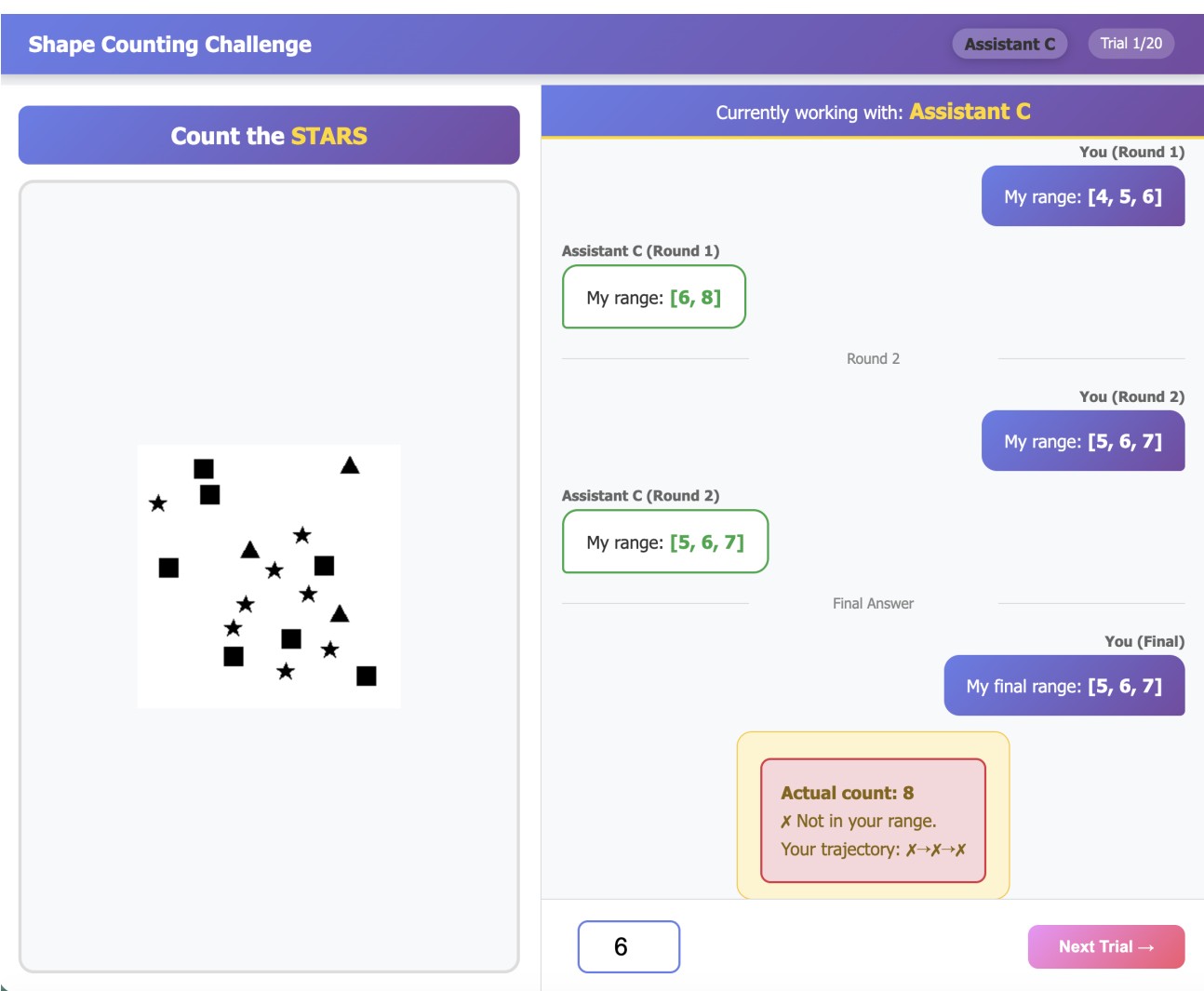

*Figure 10.* Trial completion state showing the full interaction history. The interface reveals the original image, displays both rounds of human-AI interaction (including all guesses and AI prediction sets), and indicates whether the participant's final answer contained the ground truth. This feedback screen appears after each of the 20 trials.

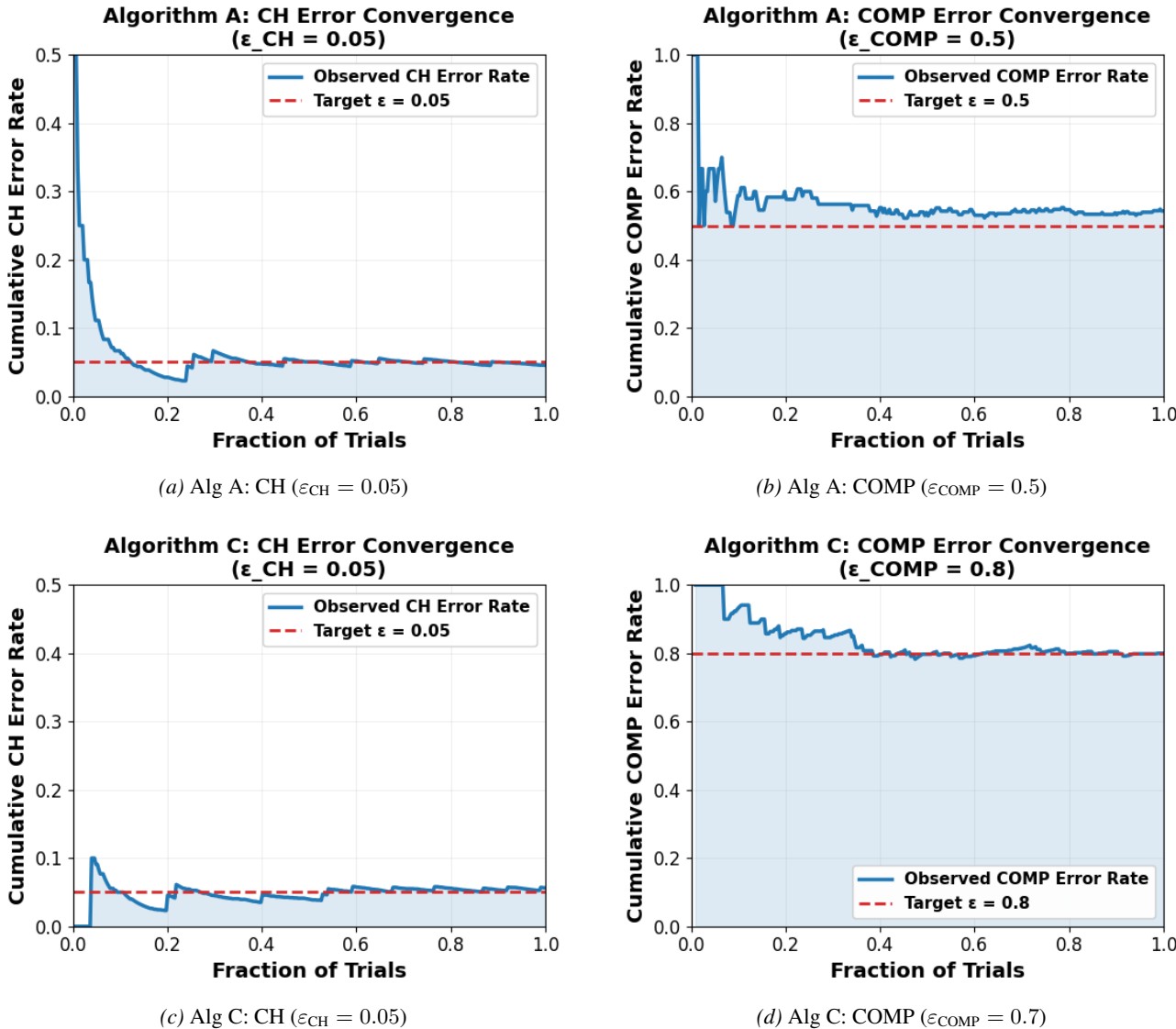

*(a)* Alg A: CH ($\varepsilon_{\text{CH}} = 0.05$)

*(b)* Alg A: COMP ($\varepsilon_{\text{COMP}} = 0.5$)

*(c)* Alg C: CH ($\varepsilon_{\text{CH}} = 0.05$)

*(d)* Alg C: COMP ($\varepsilon_{\text{COMP}} = 0.7$)

*Figure 11.* Empirical convergence for Algorithms A and C in the crowdsourcing task. All plots track the cumulative average running error, $\text{AvgError}_t = \frac{1}{t} \sum_{i=1}^{t} \mathbf{1}\{\text{Error}_i\}$, for counterfactual harm (left) and complementarity (right) as they stabilize at their respective target values.

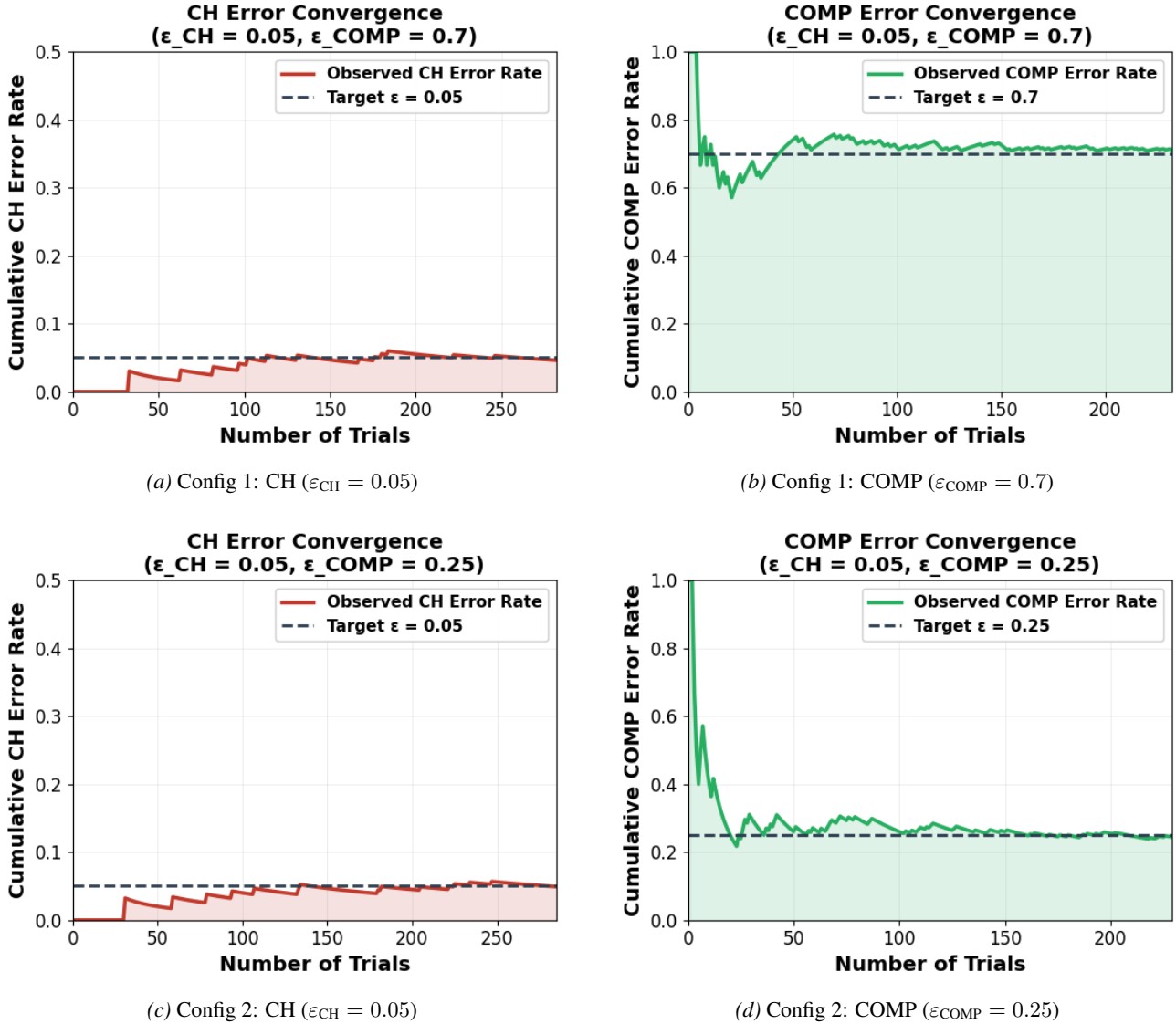

*(a)* Config 1: CH ($\varepsilon_{\text{CH}} = 0.05$)

*(b)* Config 1: COMP ($\varepsilon_{\text{COMP}} = 0.7$)

*(c)* Config 2: CH ($\varepsilon_{\text{CH}} = 0.05$)

*(d)* Config 2: COMP ($\varepsilon_{\text{COMP}} = 0.25$)

*Figure 12.* Empirical convergence in the LLM-simulated medical task across varying configurations. Plots demonstrate the algorithm's ability to track $\text{AvgError}_t = \frac{1}{t} \sum_{i=1}^{t} \mathbf{1}\{\text{Error}_i\}$ for both CH and COMP metrics under different target regimes.

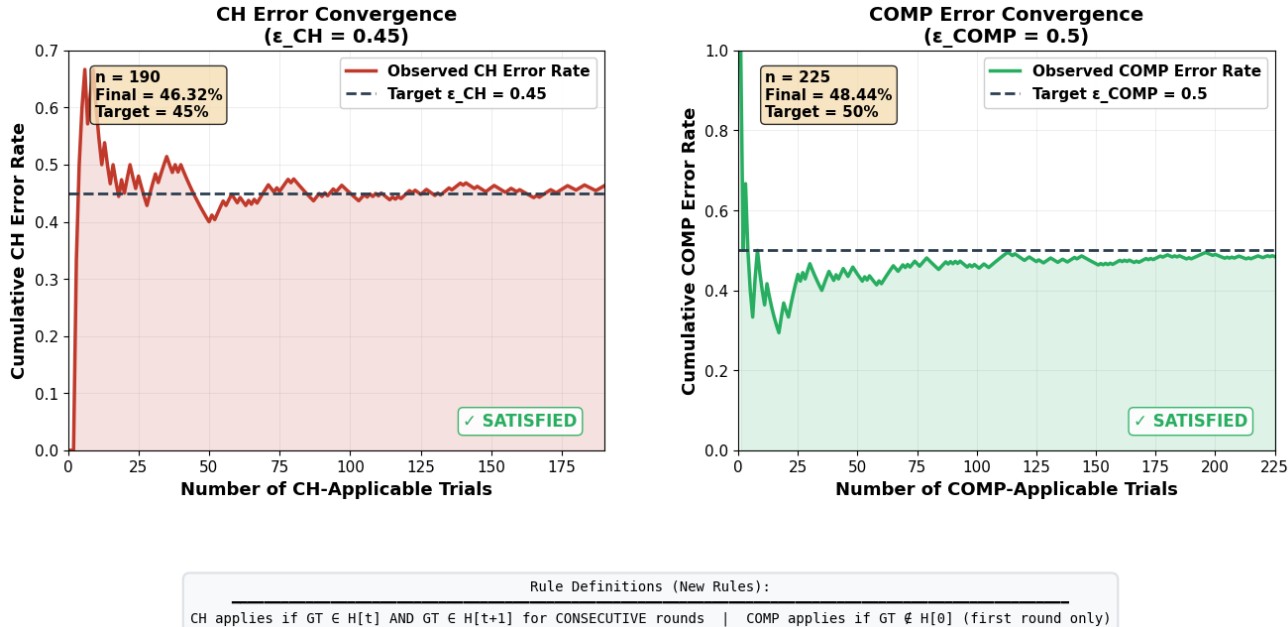

*Figure 13.* Convergence of empirical CH and Comp error rates to their targets under the alternative rule pair.

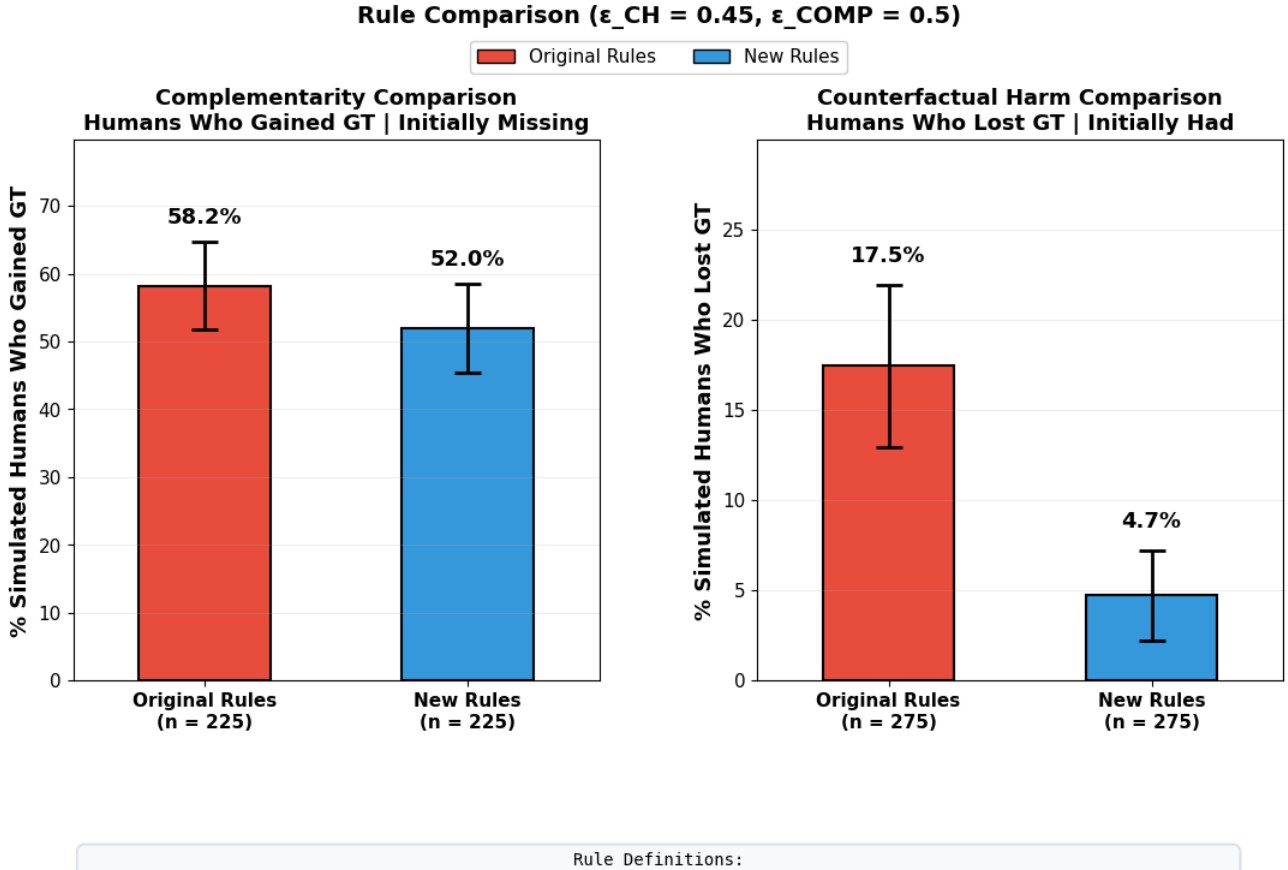

*Figure 14.* Effect of the alternative rules on human behavior: CH error tolerance on firmly-committed cases (right) and complementarity gain when controlled at the first versus final round (left).

