# OpenReview forum: "Multi-Round Human–AI Collaboration with User-Specified Requirements"
_ICML.cc/2026/Conference — ICML 2026 regular_

### Official Review · Reviewer_Mf3Z · 2026-02-23

**Soundness:** 3
**Presentation:** 4
**Significance:** 3
**Originality:** 4
**Overall Recommendation:** 5
**Confidence:** 3

**Summary:**

This paper aims to address the challenge of complex high-stakes multiturn human-AI interactions through proposing a counter-approach to the “consensus-based” deliberation previously popularised and implemented within the research community. Instead of seeking for the AI and a human to agree, the authors propose a human-centric approach to deliberation underpinned by two principles: counterfactual harm and complementarity. The authors focus on multi-turn interactions and develop an online, distribution-free algorithm which involves a user specifying harm and complementarity for a task, and the AI providing prediction sets which are updated based on daily violation thresholds. The authors validate the approach in a medical setting and a human crowdsourcing study, and find that allowing counterfactual harms to occur led humans to diverge from correct initial guesses in the medical setting, and find that the algorithm can successfully maintain target error rates during studies. As a result, the authors suggest researchers and practitioners adopt these concepts in order to improve the outcomes of human-AI assistance.

**Compliance With Llm Reviewing Policy:**

Affirmed.

**Final Justification:**

I appreciate the authors' responses to the reviews and have maintained my score.

**Key Questions For Authors:**

N/A.

**Limitations:**

Yes.

**Strengths And Weaknesses:**

**Strengths**
1. Multi-turn focus. The paper introduces an approach which specifically acknowledges that problem solving interactions are naturally multi-turn, advancing from previous static approaches to AI-collaboration dynamics.
2. Black box approach. By proposing a distribution-free approach (and treating humans as a black box) the work more accurately aligns to the way humans engage in decision-making, making the impact of this work more widely applicable.
3. Empiricism. The two experiments provided in the paper provide compelling example cases of how the two theoretical principles can be applied and the impact of them in different settings. It was particularly useful to see two quite contrasting settings deployed.

**Weaknesses**
Weaknesses
1. Generalisability to open-ended issues. The algorithm requires a fixed set of possible outcomes and structured labels. As this work attempts to push this problem towards more realistic human-scenarios, this seems like a drawback as many problems humans face will not meet this criterion as often problem spaces are open-ended.
2. Nonconformity score dependency. The utility of this work depends on the quality of the nonconformity score introduced. If the model provides impossibly long prediction sets then this could lead to humans being encumbered by this approach rather than enabled.

---

> ### Author Rebuttal · Authors · 2026-03-30
>
> Thank you for your careful reading and positive assessment. We are glad you found the multi-turn focus, black-box approach, and empirical validation compelling. We address the two weaknesses below.
>
> # On  generalizability to open-ended settings.
> This is a fair observation. Uncertainty quantification for open-ended, free-form outputs is an open challenge for the broader AI community, not specific to our collaborative framework. Even in non-collaborative settings, how to construct meaningful prediction sets for open-ended label spaces remains an active area of research. Recent work on set-valued uncertainty quantification for generative models represents important steps in this direction. In particular, Quach et al. [1] and Noorani et al. [2] provide principled means to construct prediction sets that capture infinite or unseen label spaces, and [2] can be directly applied as the CP component within our collaborative framework to mitigate this limitation. We will discuss these works in the related work section of the revised manuscript, and extending our framework to open-ended settings by building on such advances is a natural direction for future work.
>
> [1] Quach et al., "Conformal Language Modeling," arXiv:2306.10193, 2024.
>
> [2] Noorani et al., "Conformal Prediction Beyond the Seen: A Missing Mass Perspective for Uncertainty Quantification in Generative Models," arXiv:2506.05497, 2025.
>
> # On non-conformity score dependency and large prediction sets.
> This is a valid point and one that applies broadly to conformal prediction. The informativeness of the prediction set depends on the quality of the underlying nonconformity score, and poor scores can produce sets that are too large to be actionable. The conformal prediction community has developed several approaches to address this, including methods that optimize the choice of nonconformity score [3], conformal training to improve score quality [4], and post-hoc set size regularization to produce more informative sets [5]. These ideas have been studied in non-collaborative settings and bringing them into our framework is a natural extension. We also note that in our experiments, set sizes remain practically meaningful across all reported settings, suggesting that the nonconformity scores used in our evaluations do not suffer from this issue in practice. We will discuss the broader challenge and relevant mitigation strategies in the revised manuscript.
>
> [3] Angelopoulos et al., "Theoretical Foundations of Conformal Prediction," arXiv:2411.11824, 2026.
>
> [4] Stutz et al., "Learning Optimal Conformal Classifiers," arXiv:2110.09192, 2022.
>
> [5] Kiyani et al., "Length Optimization in Conformal Prediction," arXiv:2406.18814, 2024.

---

> > ### Author Rebuttal · Reviewer_Mf3Z · 2026-04-02
> >
> > Thank you for the response. I appreciate the commitment to updating the manuscript accordingly.

---

### Official Review · Reviewer_7Bd8 · 2026-03-01

**Soundness:** 2
**Presentation:** 3
**Significance:** 3
**Originality:** 2
**Overall Recommendation:** 3
**Confidence:** 3

**Summary:**

This submission claims to assess the issue of principled governance of multi-round human-AI conversational collaboration, which is increasingly used for high-stakes decision-making but lacks a rigorous framework to ensure reliable improvement of decision quality. An important concept explored by the manuscript is a human-centric collaboration paradigm governed by two core principles: counterfactual harm (ensuring AI does not undermine existing human strengths) and complementarity (ensuring AI adds value where humans are prone to errors). To formalize these principles, the authors propose a general framework that allows users to define application-specific rules for counterfactual harm and complementarity, rather than relying on a single canonical definition. They further develop an online, distribution-free algorithm with finite-sample guarantees, which can enforce user-specified constraints throughout the dynamic interaction process, without any assumptions on problem distribution, human behavior, or the AI's internal mechanism. The authors validate the framework through two complementary experimental settings: a large-scale LLM-simulated medical diagnostic task, and a human crowdsourcing study on a pictorial reasoning task. Empirical results show that the proposed algorithm maintains the prescribed violation rates of counterfactual harm and complementarity even under non-stationary interaction dynamics, and adjusting the strictness of these constraints leads to predictable shifts in human decision accuracy, confirming the two principles as practical levers to steer multi-round human-AI collaboration.

**Compliance With Llm Reviewing Policy:**

Affirmed.

**Final Justification:**

I appreciate the authors' detailed response, but it has not fully addressed all of my concerns; therefore, I will maintain my current score.

**Key Questions For Authors:**

1. The paper only tests the most basic rule instances in experiments, while Section 4.2 proposes multiple flexible user-defined rule forms. Have you conducted supplementary experiments with more complex rules (e.g., rules based on the human's multi-round proposal history)? What is the systematic impact of different rule designs on the number of collaboration rounds, human decision behavior, and final decision accuracy?
2. The paper treats the natural language dialogue between human and AI as a black box, without any constraints on the AI's textual responses. In practice, the AI's textual content may significantly alter human judgment, even if the prediction set meets the counterfactual harm and complementarity constraints. How do you view the potential impact of textual dialogue on the core theoretical guarantees? Is there any mechanism in your framework to align the textual responses with the predefined collaboration constraints?
3. The proposed algorithm uses a fixed-step update rule for threshold calibration, while the online conformal prediction field has developed various advanced strategies such as adaptive step sizes and decaying step sizes. What is the rationale for choosing the fixed-step strategy? Have you tested the performance of other step size strategies in terms of convergence speed and error control in the multi-round collaboration setting?
4. Both experiments in the paper adopt fixed information asymmetry settings, while real-world multi-round human-AI collaboration usually involves dynamically evolving information asymmetry as the dialogue proceeds. Do you think the framework can still maintain its theoretical guarantees in dynamic information asymmetry scenarios? Is there any theoretical analysis or empirical validation to support this?

**Limitations:**

yes

**Strengths And Weaknesses:**

## Strengths
1. The paper provides a complete theoretical derivation and finite-sample guarantee for the proposed online algorithm (Theorem 5.1, with detailed proof in Appendix A). The algorithm is distribution-free and requires no assumptions on the problem distribution, human behavior, or the AI's internal mechanism, which makes the theoretical guarantees robust even under non-stationary interaction dynamics, a common challenge in real-world multi-round human-AI collaboration. The core claims are fully supported by both theoretical proofs and empirical validations.
2. The paper extends the single-round human-AI collaborative prediction set framework to the more practical multi-round conversational setting, addressing the critical limitation of offline calibration methods that fail when human behavior evolves dynamically in response to AI outputs. Unlike most prior multi-round collaboration frameworks that assume symmetric agents and focus on consensus reaching, the proposed framework adopts an asymmetric, human-centric view that aligns with real-world AI deployment, where humans hold ultimate decision accountability. The user-defined rule design also enables flexible adaptation of the two core principles to diverse application scenarios, providing a novel and generalizable formulation for multi-round collaboration.
3. The authors conduct experiments in two complementary settings: a scalable LLM-simulated medical diagnosis task and a real human crowdsourcing pictorial reasoning task. The experiments not only verify that the algorithm's empirical error rates converge to the prescribed targets (consistent with the theoretical guarantees), but also provide solid evidence that counterfactual harm and complementarity are effective levers to steer human decision quality. The results directly demonstrate the practical utility of the framework, with clear implications for building safer and more effective AI decision support systems in high-stakes domains.

## Weaknesses
1. While Section 4.2 introduces a variety of feasible user-defined rules for counterfactual harm and complementarity (e.g., rules based on multi-round human insistence on the truth, or historical proposal records), the experiments in Sections 6.1 and 6.2 only adopt the most basic rule instances. The paper provides no empirical results on how more complex rule designs affect collaboration dynamics, human decision behavior, and final decision accuracy, which weakens the empirical support for the claimed generality of the user-defined rule framework.
2. In Section 4.2, the paper uses the max operator to define per-session counterfactual harm and complementarity violations (E_t^CH and E_t^Comp), but does not explain the rationale for choosing the max operator over cumulative violation counts, nor its impact on the algorithm's theoretical guarantees. In Section 5, the paper fixes the thresholds τ_t and λ_t across all rounds within a single day t, but provides no sufficient justification for this design choice, which may cause confusion for readers about the algorithm's design logic.
3. The paper treats the textual dialogue components (U_t,r and A_t,r) as black boxes, with no constraints or analysis of their impact on the core guarantees. In practice, the AI's textual responses can significantly influence human judgment, potentially leading humans to abandon correct answers even if the prediction set meets the counterfactual harm constraint. The paper does not discuss this risk, nor does it provide any mechanism to align the textual dialogue with the core constraints, creating a gap between the theoretical guarantees and real-world application.
4. Both experiments use fixed, pre-designed information asymmetry between the human and AI, while in real multi-round conversational collaboration, information asymmetry evolves dynamically as humans ask follow-up questions and share more context. The experimental setup does not simulate this dynamic information exchange, so it cannot fully validate the robustness of the framework in more realistic interaction scenarios.

---

> ### Author Rebuttal · Authors · 2026-03-30
>
> We thank the reviewer for their careful and detailed feedback. We are glad that you found the claims fully supported by theoretical and empirical validations, and our formulation of Multiround HAI collaboration novel and generalizable. We will address each of your concerns below:
>
> # On implementing more rules
> Thank you for your suggestion. We implemented two additional rules on the LLM medical diagnosis task.
> 1. **Convergence plots confirming our framework achieves the CH and COMP targets under both new rules are at https://imgur.com/a/sc3mgNq.**
> 2. **The new rules also produce intuitive effects on human behavior.**
>  For CH, the new rule applies only when the human insists on GT for two consecutive rounds. A higher CH error (45%) therefore has less negative impact: the AI is allowed to drop GT only on cases where the human is truly committed, who are less likely to be swayed by the AI's set. Under the original rule, the same error rate applies to weakly-committed cases, where the human is more likely to follow the AI and abandon GT.
> For complementarity, the original rule applies when the human lacks GT at their final round; the new rule applies when the human lacks GT at their first round. We observe lower GT gain under the new rule, which is intuitive: controlling complementarity at the last AI-provided set has a more direct impact on the human's final decision, whereas controlling it at the first round leaves room for the conversation to dilute the effect. **Summary figures: https://imgur.com/a/camNJud**
>
> # On the max operator, and the fixed thresholds across rounds.
> These are closely related and we address them together.
> The max operator reflects a deliberate modeling choice: a single rule violation within an episode is enough to potentially influence the human's subsequent reasoning, so we treat the episode as compromised from that point. This is a problem definition choice, not an algorithmic one, and it naturally captures the dynamics of human-AI conversation.
> A direct consequence is that ground truth is only revealed at episode termination, so there is no within-episode feedback to update thresholds round by round. The thresholds are therefore held fixed within an episode and updated only at the episode boundary. For applications where within-episode feedback is available, a cumulative violation count with round-by-round updates we expect would admit a straightforward modification, and we will discuss this in future work.
>
> # On textual dialogues being black boxes.
> We agree with the reviewer that the AI's textual responses can influence human judgement in arbitrary ways not captures by the prediction set guarantees. We do not claim otherwise.
> However, our framework is not designed to prevent this, it is designed to provide rigorous statistical guarantees on the sets which are far more traceable objects to reason about formally than free-form text. extending such guarantees ( CH and COMP) to natural language is an open and genuinely hard problem, and an interesting extension.
>
> Importantly, our framework is complementary to, not a replacement for, work that improved the textual quality of AI responses. Any LLM that performs well on the textual side can be used as the underlying model in our framework. our method then adds formal guarantees on the sets on top of whatever textual capabilities the model already has, without removing or degrading them. We will make this explicit in the paper and discuss it in the future work.
>
> # On dynamic information exchange
> We would like to first clarify that in both experiments, the humans and AI are free to exchange textual information, evidence and justifications across rounds. There is a genuine information flow. n the crowdsourcing study, participants view the image again each round, providing new perceptual information.
>
> That said, to further address the concern, **we ran an additional experiment on the LLM medical diagnosis task where both parties receive randomly revealed evidences at each round** and may share it freely.
> This captures two axes to dynamic information exchange: internal exchange where the human or AI can share information, and external dynamics where environment itself provides new information as conversation progresses. Our algorithm continues to control error rates under this setting. **Results: https://imgur.com/a/XwhBUvN.**
> We also note that our theoretical guarantees make no assumptions on the data distribution or information structure, so they hold regardless of how information flows across rounds.
>
> # Fixed vs adaptive set size:
> The fixed step size was chosen for theoretical convenience, yielding clean convergence guarantees. Adaptive and decaying step sizes, as explored in (Angelopoulos et al. (2024),) can accelerate convergence but may introduce instability in set sizes. The right choice depends on the application, and we will discuss this tradeoff and flag adaptive step sizes as a natural extension in the revised manuscript.

---

> > ### Author Rebuttal · Reviewer_7Bd8 · 2026-04-01
> >
> > Thank you for the detailed rebuttal. While the authors’ supplementary experiments and clarifications have partially addressed my concerns on rule design, algorithm choices, and dynamic information exchange, the core issue of aligning natural language dialogue with counterfactual harm and complementarity constraints remains unsolved. The authors treat this as an open problem, yet it creates a critical gap between the statistical guarantees of prediction sets and real human–AI collaboration, where dialogue heavily shapes human decisions. Closing this gap requires substantial theoretical and empirical extensions beyond a brief rebuttal. Thus, I will maintain my current score.

---

> > > ### Author Response · Authors · 2026-04-05
> > >
> > > We thank the reviewer for the careful follow-up and appreciate that our rebuttal addressed the concerns they explicitly highlighted regarding rule design, algorithmic choices, and dynamic information exchange. We would therefore like to briefly clarify why we do not believe the remaining concern about the textual channel diminishes the core contribution of the paper.
> > >
> > > First, the problem studied in this paper is multi-round human–AI collaboration through prediction sets. The textual components are optional and supplementary in both the interaction protocol and the algorithmic development. Our formal framework, guarantees, and algorithmic contributions are all defined in terms of the prediction sets exchanged between the human and the AI. The textual messages are included only to allow for a realistic interaction protocol and to keep the formulation general, but the guarantees themselves are about the uncertainty sets, not the free-form language channel. This is standard in the literature: existing multi-round collaboration frameworks already study meaningful interactive collaboration through structured exchanged objects such as predictions or actions, without attempting to model or constrain free-form natural language (e.g. look at [1, 2]). The absence of a theory for the text channel therefore does not make such frameworks vacuous, and likewise does not make our contribution meaningless. Rather, our paper contributes a new and principled way to study collaboration through the lens of counterfactual harm and complementarity, which, to our knowledge, did not previously exist in the multi-round setting.
> > >
> > > Second, the concern about aligning free-form dialogue with counterfactual harm and complementarity is indeed interesting, but it is a distinct and substantially broader problem. Even defining what it would mean for a natural-language response to satisfy counterfactual harm or complementarity is nontrivial, since these are statistical notions tied in our paper to whether the AI preserves or recovers the truth through its set-valued outputs. Extending such guarantees to unrestricted dialogue would therefore require new modeling choices and new methods for measuring and controlling the effect of language on human decisions. That is well beyond the scope of this paper, and, importantly, it was not part of the original claimed contribution.
> > >
> > > To this end, the contribution of the present paper should be understood as follows: we introduce and formally define multi-round human–AI collaboration through prediction sets; we extend the notions of counterfactual harm and complementarity to the multi-round setting for the first time; and we provide an online algorithm with finite-sample guarantees that enforces these user-specified requirements without making assumptions on human behavior. We view extending such guarantees to the free-form textual channel as a natural and complementary future direction, but not as a prerequisite for the present contribution to be meaningful.
> > >
> > >
> > > [1]: Tractable agreement protocols, Collina et. al.
> > >
> > > [2]: Collaborative prediction: Tractable information aggregation via agreement, Collina et. al.

---

### Official Review · Reviewer_eQma · 2026-03-02

**Soundness:** 2
**Presentation:** 3
**Significance:** 2
**Originality:** 3
**Overall Recommendation:** 3
**Confidence:** 4

**Summary:**

This paper proposes a framework for multi-round human-AI collaboration that treats the human as an accountable "black box" and governs AI behavior through two user-defined principles: Counterfactual Harm and Complementarity. The authors introduce an online, distribution-free algorithm that adjusts AI uncertainty sets to satisfy these constraints with finite-sample guarantees.

**Compliance With Llm Reviewing Policy:**

Affirmed.

**Key Questions For Authors:**

1. The core premise is that the human and AI are asymmetric agents because the human is ultimately accountable . However, prior literature suggests humans are highly susceptible to human overtrusting AI suggestions. Does the "black box" human model adequately account for scenarios where the human's "strength" is compromised by the AI's previous round output? Specifically, if the human's correct judgment is abandoned due to AI influence _before_ a CH rule is even triggered, does the framework lose its protective utility ?
2. The online calibration algorithm relies on the immediate revelation of the ground-truth label $y_t$ at the end of each interaction to update thresholds . In many high-stakes use cases mentioned, such as medical diagnostics, the "true" answer may be delayed by weeks or months. How does the algorithm maintain safety guarantees if the feedback loop is significantly delayed?
3. The experiments required human participants to maintain a fixed-size prediction set (e.g., exactly three integers) to isolate the AI's influence . In a real-world, fluid collaboration, humans naturally vary their set size based on their own changing uncertainty. How do the authors expect the performance to vary when humans are allowed to express variable-sized sets?

**Limitations:**

yes

**Strengths And Weaknesses:**

**Strengths**

This paper is strongly motivated. It convincingly argues that much of the existing literature assumes a kind of symmetry between humans and AI agents that does not really hold in practice. In real-world deployments, humans are typically the ones held accountable and retain the final say, and the paper builds on this asymmetry in a thoughtful way.

The proposed online calibration algorithm is carefully developed and comes with provable error control, and it's nice that it does not rely on overly restrictive assumptions about human rationality or specific data distributions. The evaluation is also comprehensive. I appreciated the combination of LLM simulationsand a human crowdsourcing study.

**Weaknesses**

That said, I have some reservations about the central asymmetry assumption. While it is normatively true that humans are accountable, the paper could engage more deeply with how this plays out in practice. Human decision-makers are susceptible to anchoring, automation bias, and overreliance on AI-generated scores. In such cases, the human may not function as a truly independent “anchor,” which complicates the theoretical hierarchy the framework relies on.

I am also concerned about the reliance on a predefined label space and immediate access to ground-truth feedback. Many high-stakes applications are open-ended or involve delayed outcomes (e.g., long-term medical results), which would prevent timely threshold updates. This limits the scope of tasks where the method can be directly applied.

Lastly, some experimental design choices reduce ecological validity. For example, requiring participants to maintain a fixed prediction set size helps isolate the effect of the AI’s suggestions, but it does not reflect how people naturally interact with decision aids. In real settings, individuals may flexibly adjust their level of uncertainty and the size of their candidate sets over time.

---

> ### Author Rebuttal · Authors · 2026-03-30
>
> Thank you for your thoughtful feedback. We are glad you found the framework strongly motivated, the online algorithm carefully developed, and the combination of crowdsourcing and LLM experiments compelling. We address each of the concerns and questions below.
>
> # On the asymmetry between human and AI, and susceptibility to AI influence (W1, Q1)
> We appreciate this question because it gets at the heart of our motivation. The reviewer's observation is precisely the problem our framework is designed to address. We build on two facts that together create the asymmetry. First, the human bears the ultimate responsibility for the decision. Second, and as the reviewer correctly notes, humans are known to be susceptible to AI influence. Our framework does not assume the human is an independent or infallible decision maker. Rather, it acknowledges this vulnerability and asks: given that the AI can and will influence the human, how do we constrain that influence in a principled way?
>
> Our framework addresses this through two contributions.
> 1. First, formalizing what CH and COMP even mean in a multi-round setting is itself non-trivial. Our answer is the rule-based methodology, where there is no single universal definition of harm — rather, the user specifies a rule that reflects the risk sensitivity and characteristics of their application. Examples of such rules are provided in lines 171–177. A natural and interesting follow-up question is whether there exist universal rules that are expressive enough to capture most practically relevant notions of CH and COMP. We find this a compelling direction and will discuss it in the future work section.
> 2. Second, there is an inherent tradeoff where stricter CH rules reduce the AI's freedom to be helpful, because the AI becomes more conservative to avoid any risk of negative influence. How strictly one defines CH should be calibrated to the risk tolerance of the application and the user.
> Lastly, in terms of guarantees, whatever rule the user specifies, our algorithm enforces it. On the reviewer's specific concern: what if the human abandons a correct label due to AI influence before a CH rule is triggered?. We want to address this directly: if that scenario constitutes harm in your application, it should be encoded in the rule. A CH rule is simply a formal specification of what counts as harm; if a particular form of AI influence is harmful, defining it as such in the rule means our algorithm will control it. There is no notion of harm before the rule triggers and **the rule is what defines harm in the first place.**
>
> **To better capture the effect of different rules, we conducted additional rule instantiation experiments**, which directly show that different CH rules produce measurable different effects on human behavior. For details and figures please refer to our resposne to Reviewer 78db. (https://imgur.com/a/sc3mgNq, https://imgur.com/a/camNJud)
>
> # on the delayed GT feedback ( W2, Q2 )
> This is an important point. thank you for your question. In settings where ground truth is delayed, threshold updates cannot happen until feedback arrives, and the guarantees tighten only once feedback is incorporated. A delay in feedback therefore produces a corresponding delay in the guarantees.
> That said, this is less limiting than it may appear. Our algorithm can incorporate feedback whenever it arrives, even out of order or with variable delays. Since our theoretical guarantees make no assumptions on the data distribution, a permutation in the order feedback arrives is handled naturally and it is simply another form of distribution shift that our algorithm is robust to. The guarantees accumulate over whichever episodes have received feedback, and thresholds are updated accordingly.
> Extending this to provide stronger real-time guarantees under significant delays is an interesting open problem, and prior work on online conformal prediction with delayed feedback provides a natural starting point. We will discuss this explicitly as a limitation and direction for future work in the revised manuscript.
>
> # On fixed prediction set size in the crowdsourcing experiment.
> We first note that in the LLM simulation experiments, human set sizes are not fixed and vary naturally across rounds, and we observe consistent trends there. The fixed set size in the crowdsourcing study was a deliberate design choice to isolate the effect of CH and COMP on human behavior. If participants freely varied their set size, gains or losses of ground truth could reflect shifts in human confidence rather as well as the AI influence, introducing a confounding variable that would make it harder to attribute observed effects cleanly to the constraints. Due to time constraints we cannot run the crowdsourcing with variable set sizes during the rebuttal period, but we expect similar trends to those in the LLM setting. We will conduct this experiment and include it in the camera-ready version.

---

> > ### Author Rebuttal · Reviewer_eQma · 2026-04-01
> >
> > Concerns are mostly addressed.

---

> > > ### Author Response · Authors · 2026-04-05
> > >
> > > We thank the reviewer for the follow-up and appreciate that they found the concerns to be adequately addressed. If there is any remaining concern or point that would be helpful to clarify further, we would be very happy to do so. Otherwise, we would greatly appreciate it if the reviewer would consider revising the score in light of their updated assessment.

---

### Official Review · Reviewer_GmPh · 2026-03-13

**Soundness:** 3
**Presentation:** 3
**Significance:** 2
**Originality:** 2
**Overall Recommendation:** 3
**Confidence:** 3

**Summary:**

This paper studies multi-round human–AI collaboration and proposes a framework for controlling how AI suggestions influence human decision making. The authors introduce two collaboration principles: counterfactual harm and complementarity. They formalize these principles through user-defined rules and design an online algorithm that constructs AI prediction sets while enforcing target violation rates for these constraints with finite-sample guarantees. The framework is evaluated through LLM-simulated collaboration on a medical diagnosis task and a crowdsourcing experiment on a shape counting task, where the authors analyze how controlling these constraints affects human decision outcomes.

**Compliance With Llm Reviewing Policy:**

Affirmed.

**Final Justification:**

I believe this work has clear potential in studying multi-round human–AI collaboration. However, my main concern remains that the current contribution is primarily at the level of interaction design, and the experimental focus differs from typical ICML standards. Therefore, I will maintain my score, as the work may be a better fit for other venues such as the HCI area.

**Key Questions For Authors:**

1. It would be helpful to discuss whether and how the framework can be adapted to more realistic human–LLM interaction settings.
2. Could the authors clarify the key algorithmic novelty of the proposed method? Since the framework primarily post-processes model probabilities to construct prediction sets without modifying the underlying model, it would be helpful to position the approach relative to recent conformal prediction and human–AI collaboration frameworks. What are the main methodological differences beyond applying these techniques to this interaction setting?
3. Could the authors include comparisons with prior baselines, such as existing conformal prediction and human–AI collaboration approaches? In addition, evaluating the framework using more standard ML metrics or tasks may help clarify how the proposed method improves over existing approaches.

**Limitations:**

The limitation discussion is currently quite narrow. In addition to the restriction to prediction sets, it would be helpful to discuss other limitations such as the simplified interaction protocol, the reliance on manually specified rules, the limited experimental scope, and the lack of comparison with existing human–AI collaboration methods.

**Strengths And Weaknesses:**

Strengths:
1. The paper presents an interesting perspective on human–AI collaboration, focusing on how AI suggestions influence human decisions in multi-round interaction settings.
2. The work provides a clean formalization of two collaboration properties, counterfactual harm and complementarity, and proposes a framework to control these behaviors with provable online guarantees.
3. The experiments analyze the convergence behavior of the proposed algorithm and illustrate how controlling these constraints affects human decision updates during interaction.

Weaknesses:
1. The problem setting appears simplified and somewhat outdated. The framework assumes that the human provides a candidate set and the AI returns a prediction set. However, in modern LLM usage, humans typically ask natural language questions and models generate free-form text responses. It is unclear whether this interaction protocol realistically reflects current human–LLM workflows.
2. The method does not modify or train the LLM. Instead, it wraps external rules around the model output to control the prediction set shown to humans. This feels closer to user interaction design rather than algorithmic innovation. The theoretical guarantee also appears relatively simple, relying mainly on a conventional online learning update mechanism.
3. The experimental design is also unusual compared with typical ICML evaluations. The experiments mainly analyze human behavior changes rather than ML performance improvements. In addition, the paper does not compare with prior baselines, such as existing human–AI collaboration frameworks or conformal prediction methods. It is therefore unclear how the proposed approach relates to or improves upon existing work.

---

> ### Author Rebuttal · Authors · 2026-03-30
>
> We thank the reviewer for their thoughtful feedback. We are glad you found our perspective to multiround collaboration interesting and the formulation clean and compelling.
> # On the interaction protocol.
> We agree that our framework is not meant to capture all modern human–LLM interactions. Our goal instead is to study a class of collaborative settings where prediction sets are a meaningful way to communicate uncertainty and where one wants formal guarantees on that interaction.
>
> First, prediction sets are a natural medium for communicating uncertainty in many high-stakes applications. For example, in medical diagnosis, a clinician often begins with a candidate set of plausible conditions after reviewing symptoms and context, and then uses additional tests or tools to refine that set. In such cases, a human-proposed set and an AI-refined set provide a natural collaboration protocol. The same applies to other decision-support settings where people reason through a shortlist of plausible options rather than a single label.
>
> Second, our protocol is best understood as a generalization of ordinary chatbot interaction, not a replacement for it. In a standard chatbot, the human and AI exchange free-form text. Our framework extends this by allowing both parties to additionally exchange prediction sets that carry formal guarantees on counterfactual harm and complementarity. From this perspective, we are not restricting how humans and AI communicate rather we are adding a structured, guaranteed layer on top of it.
> # On wrapper vs trained model
> This is intentional, and we view it as a strength rather than a weakness. An increasing number of frontier language models are not accessible for fine-tuning, either due to provider policies or the prohibitive cost of training at scale. A framework that works with any black-box LLM, without requiring access to model parameters, is directly deployable in the settings where practitioners actually operate.
>
> Additionally, fine-tuning a model on a specific task can degrade its performance on other tasks, aka catastrophic forgetting, and may undermine safety and alignment properties. We leave the base model untouched and provide guarantees on top of any underlying model.
> # On contributions
> We will gladly clarify our theoretical contributions. The core technical challenge in this work is mathematically formalizing what it means for a multi-round human-AI interaction to satisfy counterfactual harm and complementarity constraints. Rigorously defining these notions in the multi-turn setting, and showing how to maintain joint control over both across an arbitrary stream, is the central conceptual contribution of the paper. The resulting algorithm is clean precisely because the formulation is tight; the complexity lies in establishing the right definitions and proving that the guarantees carry through the multi-round structure, not in the mechanics of the update rule itself.
> # On experimental design and baselines
> To address this concern directly, **we conducted an additional experiment comparing our method against a carefully designed version of Adaptive Conformal Inference (ACI)**, a standard online conformal prediction method that we apply to the multi-round setting. ACI maintains marginal coverage, but does not explicitly control rule based notions of counterfactual harm or complementarity.
> The experiment is designed to show two things.
> 1. Maintaining marginal coverage alone is insufficient metric. ACI produces substantial counterfactual harm, meaning it frequently excludes labels the human correctly proposed, which actively misleads the expert over the course of the conversation.
> 2. By directly controlling counterfactual harm and complementarity, our method incorporates the human's knowledge into the prediction set, and when the human is sufficiently informative, this allows us to achieve the same marginal coverage as ACI with meaningfully smaller sets.
>
> Both points are visible in the figure at the anonymous link below, where at matched marginal coverage levels our algorithm produces smaller sets and dramatically lower counterfactual harm than ACI. We will include this experiment with additional details across the different tasks in the revised manuscript.
> **Link: https://imgur.com/a/HcsMpCg**
>
> To the best of our knowledge, no prior work addresses multi-round human-AI collaboration through the joint lens of CH and complementarity with formal guarantees, which makes direct comparisons difficult. Works on learning to defer allocate decisions between humans and models but do not produce calibrated joint prediction sets. Works on prediction sets as advice to humans optimize downstream human accuracy under a behavioral model, which our framework deliberately avoids. The human-only and AI-only baselines in our experiments, together with the new ACI comparison, represent the most meaningful points of comparison available. We will make this clearer in the revised related work section.

---

> > ### Author Rebuttal · Reviewer_GmPh · 2026-04-04
> >
> > Thank you for the detailed clarifications and additional experiments. I believe this work has clear potential in studying multi-round human–AI collaboration. However, my main concern remains that the current contribution is primarily at the level of interaction design, and the experimental focus differs from typical ICML standards. Therefore, I will maintain my score, as the work may be a better fit for other venues such as the HCI area.

---

> > > ### Author Response · Authors · 2026-04-05
> > >
> > > We thank the reviewer for the careful follow-up and appreciate their acknowledgement that the paper has clear potential for studying multi-round human–AI collaboration. We would, however, like to clarify why we do not think the remaining concern changes the fit or significance of the contribution.
> > >
> > > First, the contribution of the paper is not merely an interface design layered on top of an LLM. The paper introduces a new formal framework for multi-round human–AI collaboration through prediction sets, identifies counterfactual harm and complementarity as two core principles for evaluating such collaboration, and gives an online algorithm with finite-sample guarantees for enforcing user-specified requirements of this form. In that sense, the contribution is algorithmic and conceptual: it introduces a new learning problem, formalizes the right statistical objects, and develops a provable method for controlling them online. We believe this is well within the scope of ICML, where many papers contribute by formulating new ML problems and developing the corresponding theory and algorithms, even when the ultimate goal is not to train a new foundation model.
> > >
> > > Second, we do not view the prediction-set protocol as outdated or unrealistic. Rather, it is a deliberate abstraction that isolates the uncertainty-communication layer of collaboration. This is also consistent with a broader literature on human–AI and multi-round collaboration, where meaningful interaction is often studied through structured exchanged objects such as predictions, actions, or beliefs, rather than unrestricted free-form text (e.g. look at [1, 2]). Our paper contributes a new way of thinking about such collaboration: not through agreement or raw task accuracy alone, but through the lenses of counterfactual harm and complementarity, which, to our knowledge, had not previously been formalized in the multi-round setting.
> > >
> > > Third, the experimental design follows naturally from the contribution of the paper, and in direct response to the reviewer’s concern about baselines, **we added a new comparison to a carefully adapted version of Adaptive Conformal Inference (ACI)** (reported in our rebuttal), which is the closest existing online conformal prediction baseline we were able to identify for this setting. ACI controls marginal coverage, but it does not explicitly account for rule-based notions of counterfactual harm or complementarity. This comparison helps illustrate exactly why our framework is needed: maintaining marginal coverage alone is insufficient in multi-round human–AI collaboration, since a method like ACI can still incur substantial counterfactual harm by excluding labels the human correctly proposed. By contrast, our method directly controls these collaboration-specific quantities, and when the human is informative, this lets it achieve the same marginal coverage with meaningfully smaller sets. We believe this is the most meaningful baseline available from existing conformal prediction methods. More broadly, to the best of our knowledge, there is no prior method that addresses multi-round human–AI collaboration through the joint lens of counterfactual harm and complementarity with formal guarantees, which makes direct one-to-one comparisons inherently limited. If the reviewer is aware of a more direct baseline that fits this setting and objective, we would be very happy to include it in the revision.
> > >
> > > We therefore believe the paper makes a meaningful ML contribution: it introduces a new formalism for multi-round human–AI collaboration, develops an online algorithm with guarantees, and empirically validates that its governing principles have the intended effect on the collaboration. While the perspective is admittedly different from standard benchmark-driven evaluations, we do not believe that makes it outside the scope of ICML; rather, it places the paper among works that open up new problem settings and evaluation paradigms for machine learning.
> > >
> > > [1]: Tractable agreement protocols, Collina et. al.
> > >
> > > [2]: Collaborative prediction: Tractable information aggregation via agreement, Collina et. al.

---

### Decision · Program_Chairs · 2026-04-30

**Decision:**

Accept (regular)

**Comment:**

This paper introduces a principled framework for multi-round human–AI collaboration, formalizing counterfactual harm and complementarity as user-defined constraints enforced via an online, distribution-free algorithm with finite-sample guarantees. The formulation is clean, the asymmetric human-centric perspective is well-motivated, and the combination of LLM-simulated and crowdsourcing experiments provides complementary validation.
Reviewer opinions were mixed but ultimately lean positive. Reviewer Mf3Z recommended accept and found the originality and presentation excellent; Reviewer eQma marked concerns as fully resolved after the rebuttal. The two weak-reject reviewers raised legitimate concerns—particularly about the gap between prediction-set guarantees and the unconstrained textual dialogue channel, and the simplified interaction protocol—but these are acknowledged limitations rather than fundamental flaws. The rebuttal was substantive: it added an ACI baseline comparison, demonstrated control under additional rule designs, and provided dynamic information-exchange experiments. The authors are transparent that aligning free-form language with formal collaboration constraints is an open problem beyond this paper's scope, which is a reasonable position.
On balance, the theoretical contribution is sound, the empirical validation is adequate, and the paper opens a useful new problem formulation for the community. Accept is recommended, contingent on incorporating the rebuttal experiments and expanding the limitations discussion in the final version.